# *Legionella pneumophila* regulates host cell motility by targeting Phldb2 with a 14-3-3ζ-dependent protease effector

**Lei Song[1]\*[†], Jingjing Luo[1][†], Hongou Wang[2], Dan Huang[1], Yunhao Tan[3], Yao Liu[3], Yingwu Wang[4], Kaiwen Yu[2], Yong Zhang[1], Xiaoyun Liu[2], Dan Li[1]\*, Zhao-Qing Luo[3]\***

[1]Department of Respiratory Medicine, Center for Pathogen Biology and Infectious Diseases, Key Laboratory of Organ Regeneration and Transplantation of the Ministry of Education, State Key Laboratory for Zoonotic Diseases, The First Hospital of Jilin University, Changchun, China; [2]Department of Microbiology, School of Basic Medical Sciences, Peking University Health Science Center, Beijing, China; [3]Department of Biological Sciences, Purdue University, West Lafayette, United States; [4]School of Life Science, Key Laboratory for Molecular Enzymology and Engineering of the Ministry of Education, Jilin University, Changchun, China

**Abstract** The cytoskeleton network of eukaryotic cells is essential for diverse cellular processes, including vesicle trafficking, cell motility, and immunity, thus is a common target for bacterial virulence factors. A number of effectors from the bacterial pathogen *Legionella pneumophila* have been shown to modulate the function of host actin cytoskeleton to construct the *Legionella*-containing vacuole (LCV) permissive for its intracellular replication. In this study, we found that the Dot/Icm effector Lem8 (Lpg1290) is a protease whose activity is catalyzed by a Cys-His-Asp motif known to be associated with diverse biochemical activities. Intriguingly, we found that Lem8 interacts with the host regulatory protein 14-3-3 ζ , which activates its protease activity. Furthermore, Lem8 undergoes self-cleavage in a process that requires 14-3-3 ζ . We identified the Pleckstrin homology-like domain-containing protein Phldb2 involved in cytoskeleton organization as a target of Lem8 and demonstrated that Lem8 plays a role in the inhibition of host cell migration by attacking Phldb2.

**\*For correspondence:**
lsong@jlu.edu.cn (LS);
li_dan@jlu.edu.cn (DL);
luoz@purdue.edu (Z-QingL)

[†]These authors contributed equally to this work

**Competing interest:** The authors declare that no competing interests exist.

## Editor's evaluation

This manuscript provides new insight into the function of a Legionella pneumophila effector protein and indicate a novel mechanism by which the bacterial effector Lem8 interferes with host cell motility. These findings will be of broad interest to those in the bacterial pathogenesis and cytoskeleton fields, and represent an exciting advance in our understanding of host-pathogen interactions.

## Introduction

*Legionella pneumophila* is a Gram-negative intracellular bacterial pathogen ubiquitously found in freshwater habitats, where it replicates in a wide range of amoebae (*Richards et al., 2013*). It is believed that these natural hosts serve as the main replication niches for *L. pneumophila* in the environment and provide the primary evolutionary pressure for the acquisition and maintenance of virulence factors necessary for its intracellular lifecycle. Infection of humans by *L. pneumophila* occurs when susceptible individuals inhale aerosols generated from contaminated water, which introduces the bacterium to the lungs where it is phagocytosed by alveolar macrophages. Instead of being digested and cleared, internalized bacteria replicate within a membrane-bound compartment termed

*Legionella*-containing vacuole (LCV), leading to the development of Legionnaires' disease, a form of severe pneumonia (*Cunha et al., 2016*).

One feature associated with the LCV is its ability to evade fusion with the lysosomal network in the early phase (<8 hr postinfection in mouse bone marrow-derived macrophages [BMDMs]) of its development and the quick acquisition of proteins originating from the endoplasmic reticulum (ER) (*Kagan and Roy, 2002*; *Sturgill-Koszycki and Swanson, 2000*; *Swanson and Isberg, 1995*). Biogenesis of the LCV requires the Dot/Icm type IV secretion system that injects more than 300 effector proteins into host cells (*Qiu and Luo, 2017*). These effectors function to modulate a wide cohort of host processes, including vesicle trafficking (*Tan et al., 2011*), protein synthesis (*Shen et al., 2009*), lipid metabolism (*Gaspar and Machner, 2014*), and autophagy (*Choy et al., 2012*) by diverse biochemical mechanisms. Coordinated activity of these effectors leads to the formation of the LCV which largely resembles the ER in its morphology and protein composition (*Qiu and Luo, 2017*).

The cytoskeleton of eukaryotic cells is composed of microfilaments derived from actin polymers, intermediate filaments and microtubules, which play distinct roles in maintaining cell shape, migration, endocytosis, intracellular transport, and the association of cells with the extracellular matrix and cell–cell interactions (*Jones et al., 2019*). Due to its essential role in these cellular processes, components of the cytoskeleton, particularly the actin cytoskeleton is a common target for infectious agents. For example, *Salmonella enterica* Typhimurium utilizes a set of type III effectors, including SipC, SopE, and SptP to reversibly regulate the rearrangement of the host actin cytoskeleton to facilitate its entry into nonphagocytic cells (*Kubori and Galán, 2003*). Other bacterial pathogens such as *Chlamydia*, *Orientia tsutsugamushi*, and *Listeria* also exploit the actin cytoskeleton and microtubule networks to promote their movement in the cytoplasm of host cells and cell to cell spread (*Cheng et al., 2018*; *Grieshaber et al., 2003*; *Kim et al., 2001*).

Growing evidence indicates that manipulation of the actin cytoskeleton dynamics plays an important role in the intracellular lifecycle of *L. pneumophila*. It has been documented that chemical interference of the actin cytoskeleton structure impedes bacterial entry and replication (*Charpentier et al., 2009*). A number of Dot/Icm effectors have been shown to impose complex modulation of the host actin cytoskeleton. Among these, VipA promotes actin polymerization by functioning as a nucleator (*Franco et al., 2012*). LegK2 appears to inhibit actin nucleation by phosphorylating the Arp2/3 complex (*Michard et al., 2015*). The protein phosphatase WipA participate in this regulation by dephosphorylating several proteins involved in actin polymerization, including N-WASP, NCK1, ARP3, and ACK1, leading to dysregulation of actin polymerization (*He et al., 2019*). RavK is a metalloprotease that cleaves actin in host cells, abolishing its ability to form polymers (*Liu et al., 2017*). Ceg14 also appears to inhibit actin polymerization by a yet unrecognized mechanism (*Guo et al., 2014*). Interestingly, LegG1 has been demonstrated to promote microtubule polymerization and host cell migration by functioning as a guanine nucleotide exchange factor for the Ran GTPase (*Rothmeier et al., 2013*; *Simon et al., 2014*). Counterintuitive to the role of LegG1, cells infected by *L. pneumophila* display defects in migration in a way that requires a functional Dot/Icm system (*Simon et al., 2014*), suggesting the existence of effectors that function to block cell migration.

Herein, we demonstrate that the *L. pneumophila* effector Lem8 (Lpg1290) (*Burstein et al., 2009*) is a cysteine protease that functions to inhibit host cell migration by targeting the microtubule-associated protein Phldb2 via a mechanism that requires the regulatory protein 14-3-3 $\zeta$.

## Results

### Lem8 is a *Legionella* effector with putative cysteine protease activities

One major challenge in the study of bacterial effectors is their unique primary sequences that share little similarity with proteins of known function. Bioinformatics analysis has been proven useful in the identification of putative cryptic functional motifs embedded in their structures. We used PSI-BLAST to analyze a library of the *L. pneumophila* Dot/Icm effectors (*Zhu et al., 2011*) and found that Lem8 harbors a putative $Cys_{280}$–$His_{391}$–$Asp_{412}$ catalytic triad present in a variety of cysteine proteases (*Figure 1A*). Further analysis by HHpred (*Söding et al., 2005*) revealed that Lem8 has high probability to have structural similarity with HopN1 and AvrPphB from *Pseudomonas syringae* (*Rodríguez-Herva et al., 2012*; *Shao et al., 2002*), as well as YopT from *Yersinia enterocolitica* (*Shao et al., 2002*) and PfhB1 from *Pasteurella multocida* (*Shao et al., 2002*; *Figure 1—figure supplement 1*).

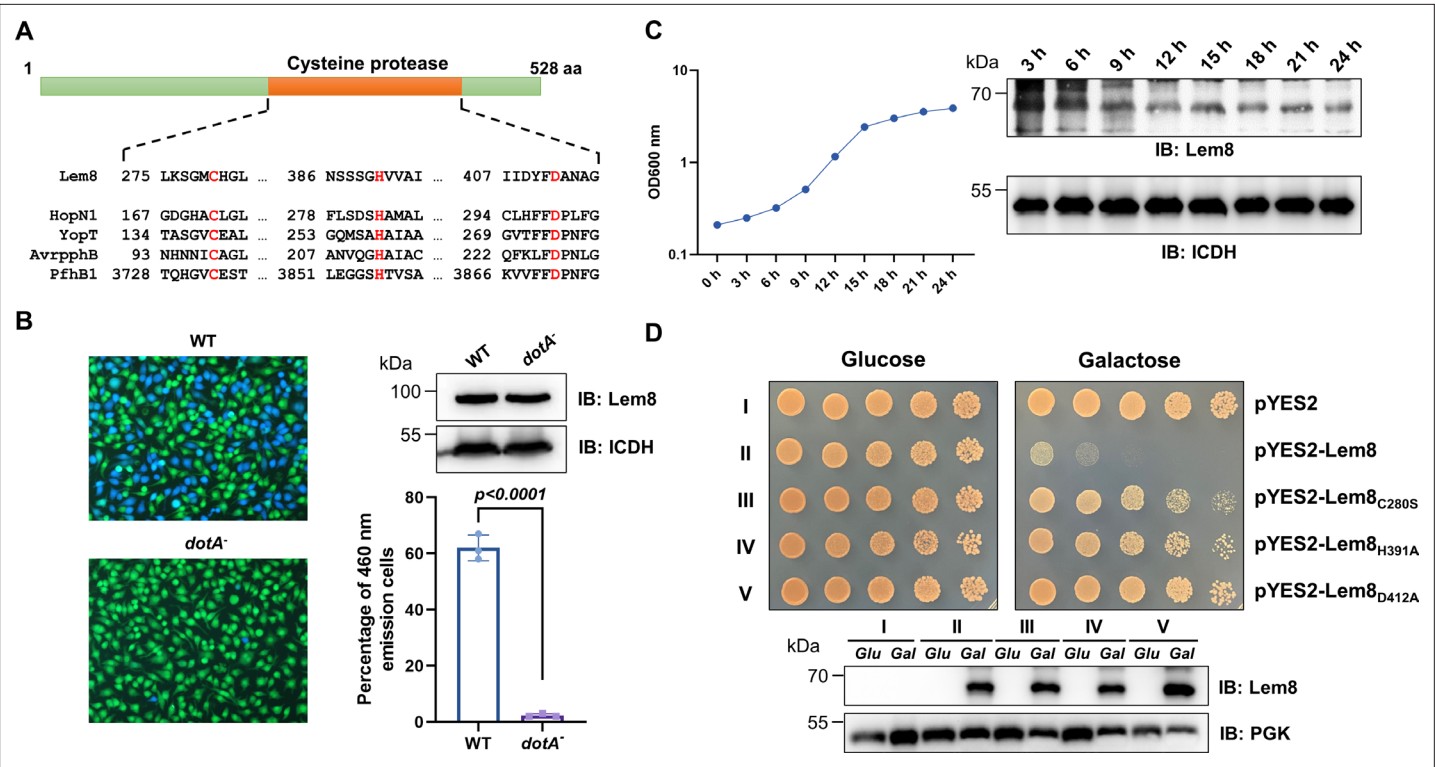

**Figure 1.** Lem8 is a cysteine protease-like Dot/Icm effector toxic to yeast. (**A**) Alignment of Lem8 with several known cysteine proteases obtained by PSI-BLAST analysis. The strictly conserved catalytic residues are marked in red. Shown cysteine proteases are HopN1 and AvrpphB from *P. syringae*, YopT from *Y. enterocolitica*, and PfhB1 from *P. multocida*. (**B**) Lem8 is translocated into mammalian cells via the Dot/Icm transporter. U937 cells were infected with wild-type *L. pneumophila* or the *dotA⁻* mutant expressing the β-lactamase-Lem8 fusion. One hour after infection, the CCF4-AM fluorescence substrate was added into the cultures and the cells were incubated for another 2 hr at room temperature before image acquisition. Cells emitting blue fluorescence signals were quantitated by counting at least 500 cells in each experiment done in triplicate. Results shown are mean ± standard error (SE) from one representative experiment. (**C**) Expression profile of *lem8* in *L. pneumophila* grown in AYE broth supplemented with thymidine. Bacteria grown to stationary phase were diluted at 1:20 in fresh medium and subcultures were grown in a shaker. Bacterial growth was monitored by measuring $OD_{600}$ at the indicated time points. Equal amounts of bacterial cells were lysed for measurement of Lem8 levels by immunoblotting with Lem8-specific antibodies. The metabolic protein isocitrate dehydrogenase (ICDH) was probed as loading control. (**D**) Lem8 is toxic to yeast in a manner that requires the predicted Cys-His-Asp motif. Yeast strains expressing Lem8 or the indicated mutants from the galactose-inducible promotor were serially diluted and spotted on the indicated media. The plates were incubated at 30°C for 48 hr before image acquisition. The expression of Lem8 and its mutants induced by galactose were determined by immunoblotting with Lem8-specific antibodies. The 3-phosphoglycerate kinase (PGK) was detected as loading control.

The online version of this article includes the following figure supplement(s) for figure 1:

**Figure supplement 1.** Sequence alignment of Lem8 with four bacterial cysteine protease effectors.

**Figure supplement 2.** Lem8 is dispensable for intracellular growth of *L. pneumophila*.

Lem8 is a protein of 528 residues coded for by the gene *lpg1290* in *L. pneumophila* strain Philadelphia 1, it was first identified as a substrate of the Dot/Icm transporter by a machine learning approach (*Burstein et al., 2009*). The translocation of Lem8 by the Dot/Icm system into host cells during *L. pneumophila* infection was later validated by two independent reporter systems (*Huang et al., 2011*; *Zhu et al., 2011*). Consistent with these results, we observed Dot/Icm-dependent translocation of Lem8 into host cells using the β-lactamase- and CCF2-based reporter assay. Approximately 60% of the cells infected with a Dot/Icm-competent strain expressing the β-lactamase-fusion emitted blue fluorescence signals. No translocation was detected when the same fusion was expressed in the *dotA⁻* mutant defective in the Dot/Icm system (*Berger and Isberg, 1993*; *Figure 1B*).

The expression of many Dot/Icm substrates peaks in the postexponential phase, probably due to the demand for high quantity of effectors to thwart host defense in the initial phase of LCV construction (*Luo and Isberg, 2004*; *Segal, 2013*). Thus, we evaluated the expression pattern of *lem8* throughout

the entire growth cycle of *L. pneumophila* in broth. Intriguingly, unlike most of effectors, the expression of *lem8* is detected at high levels in the lag phase of its growth cycle in bacteriological medium. A decrease in protein abundance is detected 9 hr after the subcultures have started and is maintained constant throughout the remaining 15-hr experimental duration (*Figure 1C*). These results suggest that Lem8 may play a role in the entire intracellular lifecycle of *L. pneumophila*.

Next, we attempted to determine whether the putative cysteine protease motif is important for the effects of Lem8 on eukaryotic cells. We first tested whether Lem8 is toxic to yeast and if so, whether the $Cys_{280}$–$His_{391}$–$Asp_{412}$ motif is required for such toxicity. Expression of Lem8 from the galactose-inducibe promoter caused cell growth arrest (*Figure 1D*). Mutations in $Cys_{280}$, $His_{391}$, or $Asp_{412}$ did not affect the stability of the protein in yeast, but abolished such toxicity (*Figure 1D*). Thus, the putative cysteine protease activity conferred by the predicted $Cys_{280}$–$His_{391}$–$Asp_{412}$ catalytic triad very likely is important for the effects of Lem8 on eukaryotic cells.

Genomic analysis reveals that Lem8 orthologs are present in several different isolates of *L. pneumophila*, including serogroup one strains Thunder Bay (*Schoch et al., 2020*), LPE509 (*Ma et al., 2013*), Paris (*Cazalet et al., 2004*), HL06041035 (*Gomez-Valero et al., 2011*), and strain ATCC 43290 (serogroup 12) (*Gomez-Valero et al., 2011*). In addition, a *lem8* homolog is also present in *L. waltersii*, one of the 40 *Legionella* species whose genomes had been fully sequenced (*Burstein et al., 2016*). Such a low prevalence suggests that Lem8 plays a role in the survival of the bacteria in specific inhabits, or its role in other *Legionella* species is substituted by genes of little sequence similarity that may have arisen by convergent evolution. We probed the role of *lem8* in *L. pneumophila* virulence by examining intracellular replication of the Δ*lem8* mutant in the protozoan host *Dictyostelium discoideum* and in BMDMs. In both host cells, intracellular growth of the Δ*lem8* mutant was indistinguishable to that of the wild-type strain (*Figure 1—figure supplement 2*), which is akin to most mutants lacking one single Dot/Icm substrate gene (*Qiu and Luo, 2017*).

## Lem8 directly interacts with the regulatory protein 14-3-3ζ

To identify the host target of Lem8, we performed a yeast two-hybrid screening against a human cDNA Library (BD Biosciences) using $Lem8_{C280S}$ fused to the DNA-binding domain of the transcriptional factor GAL4 as bait. Plasmid DNA of the library was introduced into the yeast strain PJ69-4A (*James et al., 1996*) expressing the bait fusion and colonies that appeared on selective medium were isolated and the inserts of the rescued plasmids capable of conferring the interactions were sequenced. We found that 50 out of the 93 independent clones analyzed harbored portions of the gene coding for 14-3-3 ζ, a member of a chaperone family important for the activity of a wide variety of proteins in eukaryotic cells (*Pennington et al., 2018*). Robust interactions occurred in the yeast two-hybrid system when full-length 14-3-3 ζ was fused to the AD domain of Gal4 (*Figure 2A*). The remaining clones were not pursued further as the interactions cannot be confirmed by immunoprecipitation or the proteins are common false-positive hits in yeast two-hybrid screenings and/or appeared not relevant to the lifecycle of *L. pneumophila* (*Supplementary file 1*).

We further explored the interactions between 14-3-3 ζ and Lem8 by immunoprecipitation (IP) assays. Flag-tagged 14-3-3 ζ was coexpressed with GFP-tagged Lem8 or GFP in HEK293 cells. IP using the Flag antibody specifically precipitated GFP-Lem8, whereas GFP was not detectable in similar experiments. Reciprocally, IP with GFP antibodies specifically pulled down Flag-14-3-3 ζ (*Figure 2B*). These results suggest that Lem8 forms a complex with 14-3-3 ζ in mammalian cells.

To determine whether Lem8 directly binds to 14-3-3 ζ, we purified recombinant proteins and used GST pulldown assays to analyze their interactions. We found that mixing $His_6$-Lem8 and GST-14-3-3 ζ in reactions led to the formation of stable protein complexes that can be retained by GST beads (*Figure 2C*).

Members of the 14-3-3 family commonly recognize phospho-serine and/or phospho-threonine sites of client proteins for binding (*Muslin et al., 1996*). Yet, we did not detect phosphorylation on Lem8 purified from mammalian cells or *Escherichia coli* using a pan phospho-serine/threonine antibody. As a control, this antibody detected phosphorylation on CTNNB1, an established phosphorylated target of 14-3-3 ζ (*Tian et al., 2004*).

To determine the region in Lem8 involved in binding 14-3-3 ζ, we constructed a series of Lem8 deletion mutants and examined their interactions with 14-3-3 ζ by immunoprecipitation. Whereas removing as few as 25 residues from the amino terminal end of Lem8 abolished its ability to bind

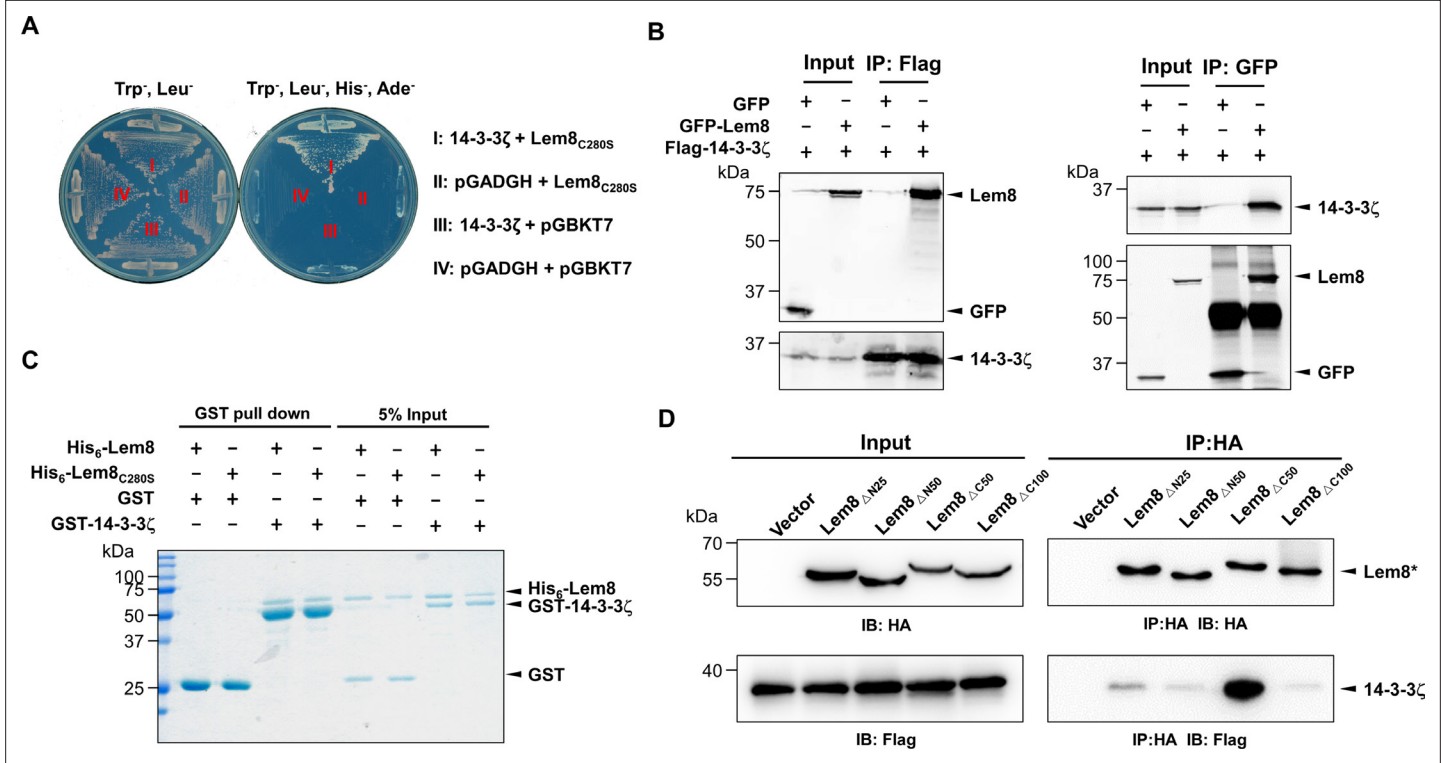

**Figure 2.** The interactions between Lem8 and 14-3-3 ζ . (**A**) Interactions between Lem8 and 14-3-3 ζ detected by yeast two-hybrid assay. Yeast strains harboring the indicated constructs were streaked on Leu⁻ and Trp⁻ medium to select for plasmids (left) or on Leu⁻, Trp⁻, Ade⁻, and His⁻ medium to assess the interactions (right). Images were acquired after 3-day incubation at 30°C. (**B**) Lem8 and 14-3-3 ζ form a protein complex in mammalian cells. Total lysates of HEK293T cells transfected with the indicated plasmid combinations were immunoprecipitated with a Flag-specific antibody (left panels) or GFP-specific antibodies (right panels), and the precipitates were probed with both Flag and GFP antibodies. Similar results were obtained from at least three independent experiments and the data shown here were from one representative experiment. (**C**) Lem8 directly interacts with 14-3-3 ζ . GST-14-3-3 ζ was incubated with His₆-Lem8 or His₆-Lem8$_{C280S}$, and the potential protein complex was captured by glutathione beads for 1 hr at 4°C. After extensive washing, bound proteins were solubilized with sodium dodecyl sulfate (SDS) loading buffer, and proteins were detected by Coomassie brilliant blue staining after being resolved by SDS/polyacrylamide gel electrophoresis PAGE. Similar results were obtained from at least three independent experiments and the data shown here were from one representative experiment. (**D**) Interactions between 14-3-3 ζ and Lem8 deletion mutants. Lysates of 293T cells expressing Flag-14-3-3 ζ and each of the HA-tagged deletion Lem8 were subjected to immunoprecipitation with the anti-HA antibody and the presence of 14-3-3 ζ in the precipitates was probed with the Flag-specific antibody. Similar results were obtained from at least three independent experiments and the data shown here were from one representative experiment.

The online version of this article includes the following figure supplement(s) for figure 2:

**Figure supplement 1.** Phosphorylation of Lem8 is not required for 14-3-3 ζ binding.

14-3-3 ζ , a Lem8 mutant lacking the last 50 residues can still robustly interact with 14-3-3 ζ , and deleting an additional 50 residues from this end abolished the binding (*Figure 2D*). Thus, either 14-3-3 ζ recognizes a large region of Lem8 or deletion from either end of this protein caused significant disruptions in its structure and abolished its ability to interact with 14-3-3 ζ .

## Lem8 undergoes 14-3-3ζ-dependent autocleavage

Since Lem8 harbors the predicted Cys$_{280}$–His$_{391}$–Asp$_{412}$ catalytic triad associated with proteases from diverse bacterial pathogens, we next investigated whether it cleaves 14-3-3 ζ . Incubation of recombinant His₆-Lem8 with His₆-14-3-3 ζ at room temperature for 2 hr did not lead to detectable cleavage of 14-3-3 ζ . Unexpectedly, a protein with a molecular weight slightly smaller than that of Lem8 was detected in this reaction (*Figure 3A*). The production of this smaller protein did not occur in reactions that contained the Lem8$_{C280S}$ mutant or when the cysteine protease-specific inhibitor E64 was included in the reactions (*Figure 3A*), suggesting that this band represents a fragment of Lem8 produced by self-cleavage. Intriguingly, the self-cleavage did not occur in samples containing only Lem8, suggesting that the self-cleavage activity of Lem8 requires 14-3-3 ζ as a cofactor.

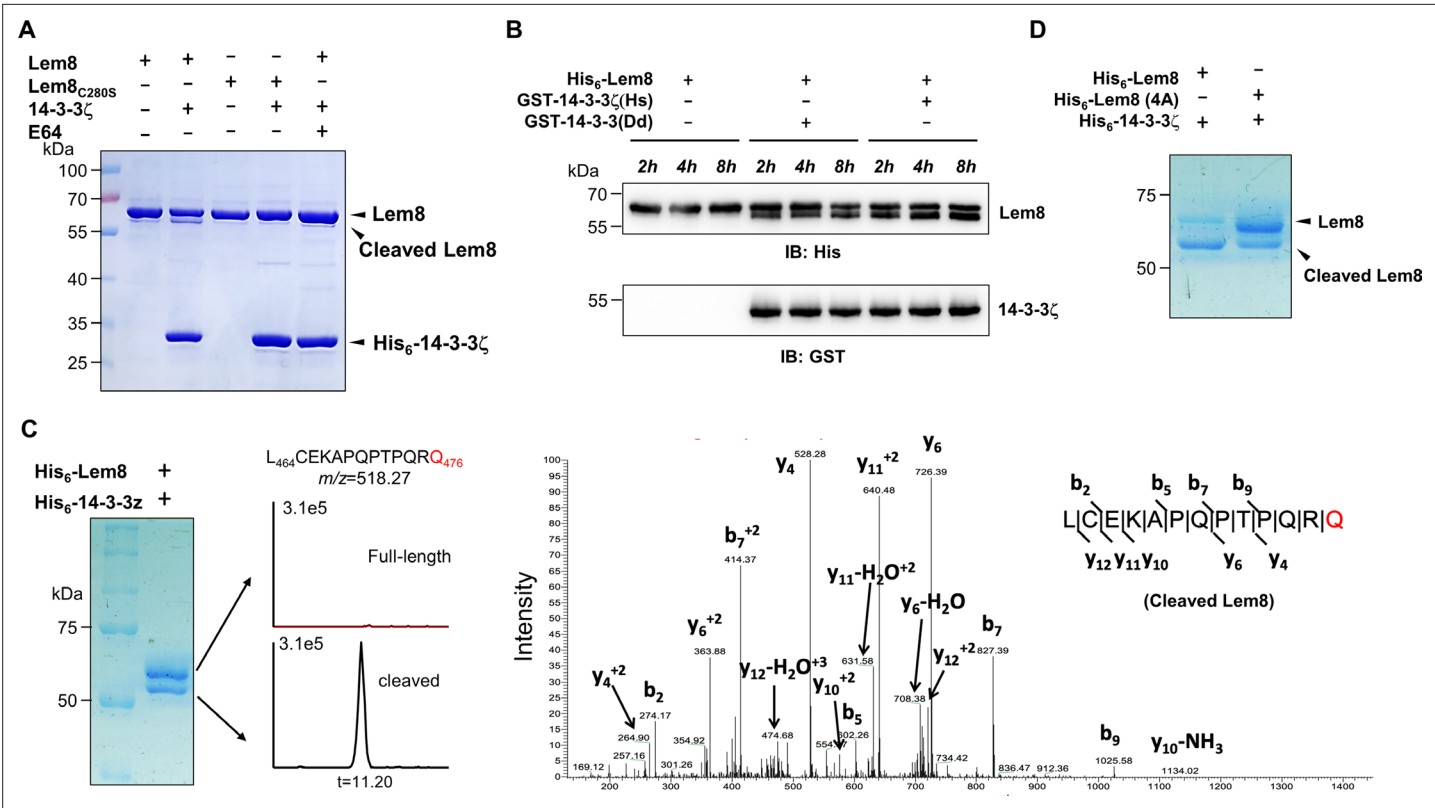

**Figure 3.** 14-3-3 ζ induces Lem8 to undergo self-cleavage. (**A**) Self-processing of Lem8 requires 14-3-3 ζ . His₆-Lem8 or His₆-Lem8_C280S was incubated with His₆-14-3-3 ζ for 2 hr, proteins resolved by sodium dodecyl sulfate–polyacrylamide gel electrophoresis (SDS–PAGE) were detected by Coomassie brilliant blue staining. The cysteine protease inhibitor E64 was added to the indicated samples at a final concentration of 10 μM. Similar results were obtained from at least three independent experiments and the data shown here were from one representative experiment. (**B**) The 14-3-3 protein from *D. discoideum* induces the self-cleavage of Lem8. His₆-Lem8 was incubated with GST-14-3-3 ζ or GST-14-3-3Dd for the indicated time and the mixtures separated by SDS–PAGE were detected by immunoblotting with antibodies specific for Lem8 and GST, respectively. Similar results were obtained from at least three independent experiments and the data shown here were from one representative experiment. (**C**) Determination of the self-cleavage site of Lem8. His₆-Lem8 was incubated with His₆-14-3-3 ζ for 16 hr, proteins were resolved by SDS–PAGE, stained with Coomassie brilliant blue. Protein bands corresponding to full-length and cleaved Lem8 band was excised, digested with trypsin and analyzed by mass spectrometry. The detection of the semitryptic peptide -L₄₆₄CEKAPQPTPQRQ₄₇₆- in cleaved samples suggested that the cleavage site lies between Gln476 and Arg477. (**D**) Mutations in cleavage site does not abolish Lem8 self-processing. Recombinant protein of Lem8 and the 4A mutant were each incubated with His₆-14-3-3 ζ for 4 hr. Proteins resolved by SDS–PAGE were detected by Coomassie brilliant blue. Similar results were obtained from at least three independent experiments and the data shown here were from one representative experiment.

The online version of this article includes the following figure supplement(s) for figure 3:

**Figure supplement 1.** Identification of the self-cleavage sites of Lem8.

*Dictyostelium discoideum*, the protozoan host of *L. pneumophila* codes for one 14-3-3 protein with 66% identity and 78% similarity to that of human 14-3-3 ζ (*Eichinger et al., 2005*), we investigated whether the *D. discoideum* 14-3-3 (14-3-3Dd) can activate Lem8 by incubating His₆-Lem8 with GST-14-3-3Dd or human 14-3-3 ζ (14-3-3 ζ Hs). In each case, we observed the production of a protein with a size clearly smaller than Lem8 as early as 2 hr after the reaction has started (*Figure 3B*). Thus, Lem8 can be activated by 14-3-3 from both humans and a protist.

To determine the self-cleavage site of Lem8, we incubated His₆-Lem8 with His₆-14-3-3 ζ at room temperature for 16 hr. Proteins resolved by sodium dodecyl sulfate–polyacrylamide gel electrophoresis (SDS–PAGE) were stained with Coomassie brilliant blue and bands corresponding to full-length and cleaved Lem8 were excised, digested with trypsin and sequenced by mass spectrometry, respectively (*Figure 3C*). To determine the potential self-cleavage site, we compared the abundance of identified tryptic peptides from full-length and cleaved Lem8 and found that the abundance of -A₄₆₈PQPTPQR₄₇₅- was similar between two sets of samples, whereas peptide -A₄₇₈QSLSAETER₄₈₇- was present only in full-length samples but not in the cleaved ones (*Figure 3—figure supplement 1A*), suggesting that

the cleavage site was between R475 and R487. Further analysis of the tryptic fragments identified a semitryptic fragment -$L_{464}$CEKAPQPTPQRQ$_{476}$- present in the cleaved protein but not in the full-length protein, indicating that the cleavage occurs between Gln476 and Arg477 (*Figure 3C*).

We further examined the self-cleavage of Lem8 by fusing GFP to the carboxyl end of Lem8, Lem8$_{C280S}$ and Lem8$_{\Delta C52}$, respectively. These fusion proteins were expressed in HEK293T cells and the cleavage was probed by immunoblotting with GFP-specific antibodies. We found that a fraction of Lem8-GFP and Lem8$_{\Delta C52}$-GFP has lost the GFP portion of the fusions, an event that did not occur in Lem8$_{C280S}$-GFP (*Figure 3—figure supplement 1B*). To determine whether Lem8 undergoes autocleavage via the recognition of the protein sequence around Gln$_{476}$, we introduced mutations to replace residues Pro$_{473}$, Gln$_{474}$, Arg$_{475}$, and Gln$_{476}$ with alanine and incubated this Lem8 mutant (called 4A) with 14-3-3 $\zeta$. Unexpectedly, although at a lower rate, self-cleavage still occurred in this mutant (*Figure 3D*). Thus, although the amino acids adjacent to Gln476 play a role in its self-cleavage, other factors such as the overall structure of Lem8 may contribute to the recognition of the cleavage site. We next analyzed the self-cleaved products of the 4A mutant and found that the cleavage site in this protein lies between Lys467 and Ala468, which is in close proximity to the one found in the wild-type protein (*Figure 3—figure supplement 1C*). Thus, the self-cleavage of Lem8 may occur at at least two sites that locate at approximately 50 residues from its carboxyl end.

## Lem8 targets Phldb2 for cleavage

It has been reported that some bacterial cysteine proteases cleave both themselves and their substrates in host cells by recognizing sites with similar sequences. For instance, AvrpphB and Avrrpt2, two type III effectors from *P. syringae* cleave themselves as well as their host targets PBS1 and RIN4, respectively (*Chisholm et al., 2005*; *Shao et al., 2003*). Importantly, in each case, the sequences of the recognition sites for both self-cleavage and cellular target cleavage are very similar. In fact, this feature has been exploited to predict the potential host substrates of these effectors by bioinformatic analyses. Therefore, we performed BLAST searches and obtained 10 candidate proteins that contain sequence elements resembling the self-cleavage site of Lem8, including Phldb2, Rasgrp2, Pak6, Exoc8, Ankrd13B, Chkb, Ppp6R1, Kiaa1033, Gnal, and Gpr61. The predicted recognition sites in these proteins locate in the middle or at sites close to either their amino or carboxyl ends (*Figure 4A*). Further experiments revealed that one of the candidates, the Pleckstrin homology-like domain family B member 2 (Phldb2) can be cleaved by Lem8 in a process that requires an intact Cys$_{280}$–His$_{391}$–Asp$_{412}$ catalytic triad. In contrast, no detectable cleavage occurred in reactions containing recombinant proteins of other candidates purified from mammalian cells (*Figure 4—figure supplement 1A*). We cannot detect the expression of Flag- or HA-GPR61 using multiple constructs and this protein was not analyzed further. The predicted Lem8 recognition site locates between the 1106th residue and the 1119th residue in this protein of 1253 amino acids (*Figure 4A*). In HEK293T cells, expression of Lem8 led to a considerable reduction of endogenous Phldb2 (*Figure 4B*). To confirm this finding, we added an HA and a Flag tag to the amino and carboxyl end of Phldb2, respectively, and coexpressed the double tagged protein in HEK293T cells with Lem8 or each of the mutants with mutations in one of the three sites (C280S, H391A, and D412A) predicted to be critical for catalysis. Detection of tagged Phldb2 by immunoblotting with the Flag-specific antibody indicated that the protein levels in cells expressing Lem8 were reduced comparing to samples in which the catalytically inactive mutants were expressed. Athough to a lesser extent, reduction in Phldb2 was also observed in experiments in which the tagged protein was detected with the HA antibody (*Figure 4C*).

Phldb2 is plasma membrane-associated protein that contains a phosphatidylinositol-3,4,5-triphosphate (PIP3)-binding domain located at its carboxyl terminus (*Paranavitane et al., 2003*), we next examined how Lem8-mediated cleavage impacts its cellular localization. In HEK293T cells, when GFP-Phldb2 was ectopically expressed, the GFP signals mainly were associated with the plasma membrane, and this pattern of distribution remains unchanged in cells coexpressing enzymatically inactive Lem8 mutants (*Figure 4D*). In contrast, in cells coexpressing wild-type Lem8, the GFP signals redistributed to occupy the entire cytoplasm, including the nuclei of transfected cells, a pattern similar to that of GFP itself (*Figure 4D*). These observations suggest that the GFP tag had been cleaved from the GFP-Phldb2 fusion to assume its typical localization in these cells. We also analyzed how Lem8 impacts the subcellular localization of endogenous Phldb2. In cells expressing mCherry-Lem8$_{C280S}$, Phldb2 is mainly associated with the plasma membrane. In contrast, in cells expressing mCherry-Lem8,

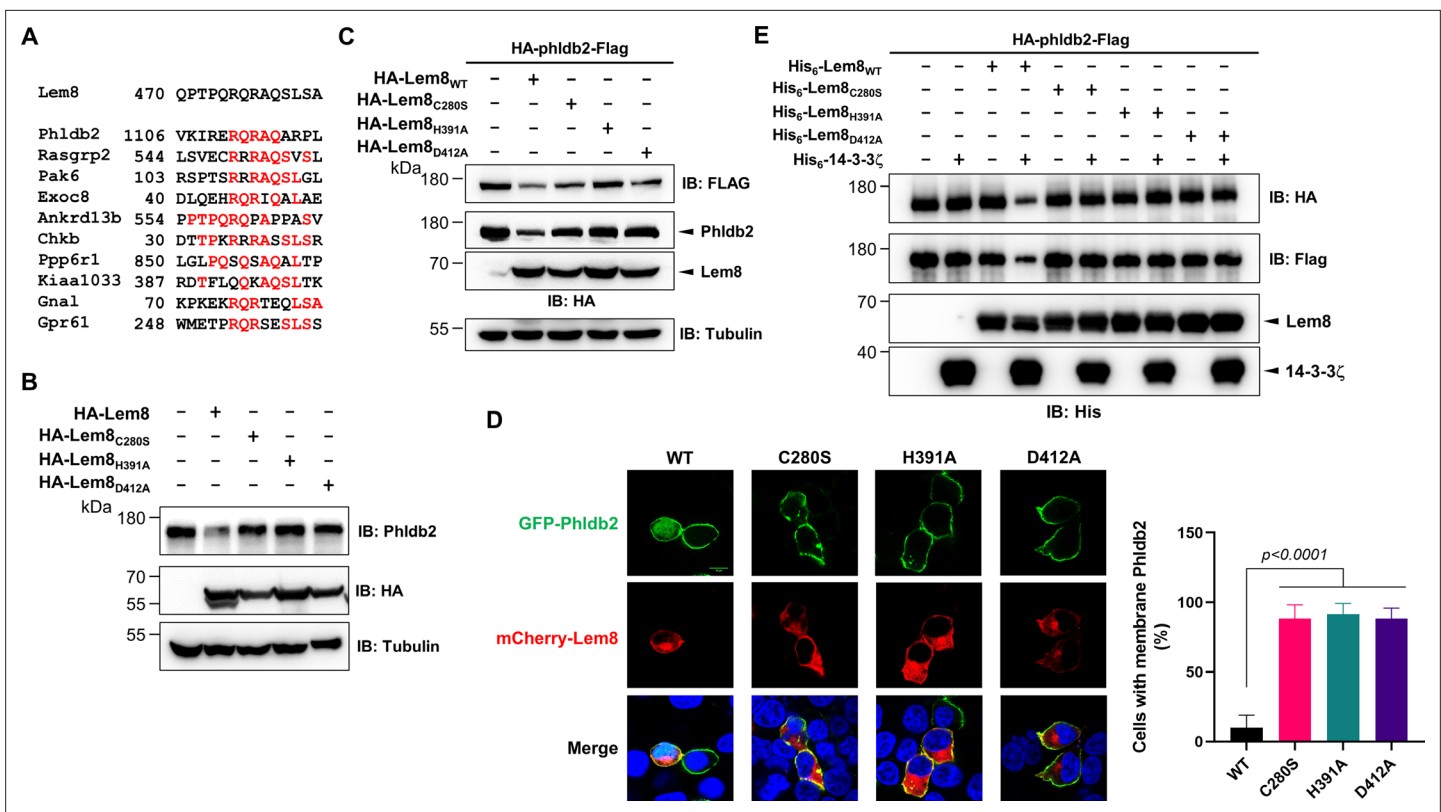

**Figure 4.** Lem8 cleaves Phldb2 in a manner that requires 14-3-3 ζ . (**A**) Multiple alignments of the self-cleavage site of Lem8 with potential targets in human cells identified by bioinformatic analysis. Identical residues are highlighted in red. (**B**) Lem8 reduces the protein levels of endogenous Phldb2 in mammalian cells. Lem8 and the indicated mutants were individually expressed in HEK293T cells by transfection. Twenty-four hours after transfection, the samples were resolved by sodium dodecyl sulfate–polyacrylamide gel electrophoresis (SDS–PAGE) and detected by immunoblotting with anti-Phldb2 antibodies. Tubulin was used as a loading control. Results shown were one representative from three independent experiments with similar results. (**C**) Lem8 cleaves exogenous Phldb2 in mammalian cells. HA and Flag tag were fused to the amino and carboxyl end of Phldb2, respectively, and the double tagged protein was coexpressed in HEK293T cells with Lem8 or each of the mutants. Twenty-four hours after transfection, the samples were resolved by SDS–PAGE and probed by a HA-specific antibody and a Flag-specific antibody, respectively. Tubulin was detected as a loading control. Results shown were one representative from three independent experiments with similar results. (**D**) Lem8 alters the subcellular distribution of GFP fused to Phldb2. GFP was fused to the amino end of Phldb2 and the protein was coexpressed in HEK293T cells with mCherry-Lem8 or each of the mutants. Twenty-four hours after transfection, cells were fixed and nucleus were stained by Hoechst 33,342. The fluorescence Images of GFP (green), mCherry (red), and Hoechst (blue) were acquired with a Zeiss LSM 880 confocal microscope. The percentage of cells with membrane Phldb2 was calculated in Phldb2- and Lem8-positive cells (right panel). Bar, 10 μm. (**E**) 14-3-3 ζ is required for the cleavage of Phldb2 by Lem8. HA-Phldb2-Flag was expressed in HEK293T cells, immunoprecipitated with a Flag-specific antibody, and eluted with 3× Flag peptides. Purified Phldb2 was incubated with His₆-Lem8 or each of the mutants in reactions with or without His₆-14-3-3 ζ . Total proteins of all samples were resolved with SDS–PAGE, and probed by immunoblotting with a HA-specific antibody, a Flag-specific antibody and a His-specific antibody. Results shown were one representative from three independent experiments with similar results.

The online version of this article includes the following figure supplement(s) for figure 4:

**Figure supplement 1.** Verification of Lem8-mediated cleavage of candidate proteins and its cleavage of phldb2 at multiple sites.

the association of Phldb2 with the plasma membrane almost became undetectable (***Figure 4—figure supplement 1B***).

We next examined whether the cleavage of Phldb2 by Lem8 occurs in a cell-free reaction. HA-Phldb2-Flag expressed in HEK293T cells isolated by immunoprecipitation was incubated with Lem8 or its inactive mutants with or without 14-3-3 ζ . Cleavage of Phldb2 occurred only in reactions containing wild-type Lem8 and 14-3-3 ζ (***Figure 4E***). Taken together, these results establish Phldb2 as a target of Lem8.

Our results using the double tagged Phldb2 suggest that Lem8 likely cleaves Phldb2 not only at the predicted site located in the carboxyl end of the protein, but also targets its amino terminal portion (***Figure 4C–E***). To test this hypothesis, we constructed two Phldb2 mutants by replacing

residues Arg$_{1111}$ and Gln$_{1112}$ (Phldb2$_{AA1}$) or Gln$_{1112}$ and Arg$_{1113}$ (Phldb2$_{AA2}$) within the predicted recognition sequence with alanine. Each of these mutants was coexpressed with Lem8 in HEK293T cells by transfection. Comparing to samples expressing enzymatically inactive Lem8, the amounts of protein detected by the amino terminal Flag epitope and the carboxyl end HA tag both decreased in cells coexpressing wild-type Lem8 (*Figure 4—figure supplement 1B*). We validated this conclusion by making constructs in which GFP was fused to the amino terminal end of Phldb2 and three of its truncation mutants, Phldb2$_{\triangle N50}$, Phldb2$_{\triangle N100}$, and Phldb2$_{\triangle N200}$, respectively. Each of these fusion proteins was coexpressed with Lem8 or Lem8$_{C280S}$ in HEK293T cells and the protein level of these fusions was probed by immunoblotting with GFP-specific antibodies. In each case, a fraction of the protein has lost the GFP portion of the fusions when coexpressed with Lem8 but not with Lem8$_{C280S}$ (*Figure 4—figure supplement 1C*). Intriguingly, the cleavage also occurred in fusion proteins in which the GFP is fused to the carboxyl terminus of Phldb2$_{\triangle C100}$ or Phldb2$_{\triangle C153}$ (*Figure 4—figure supplement 1D*, right panel). These results are consistent with the notion that Lem8 targets Phldb2 at multiple sites.

We also attempted to detect Lem8-mediated cleavage of endogenous Phldb2 in cells infected with *L. pneumophila*. Although Lem8 translocated into infected cells by a Dot/Icm-competent strain expressing Lem8 from a multicopy plasmid is readily detetable, we were unable to detect Phldb2 cleavage in these samples (*Figure 4—figure supplement 1E*). The most likely reason for the inability

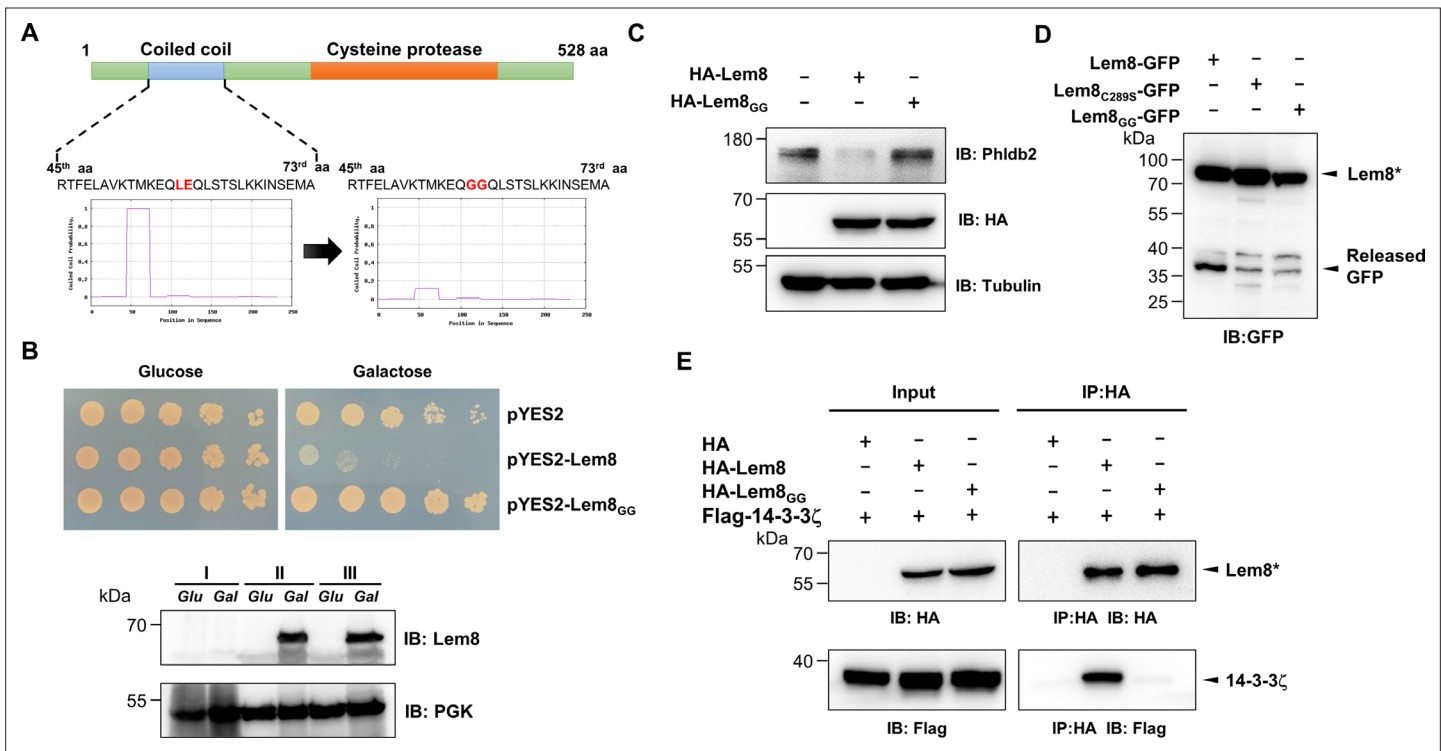

**Figure 5.** A coiled coil motif in Lem8 is important for its interactions with 14-3-3 ζ . (**A**) Lem8 harbors a putative coil motif. A predicted coiled coil motif located in the amino end of Lem8 (top panel). The sequence ranges from the 45th residue to the 73rd residue with a coiled coil probability of 100% according to MARCOIL (lower panel, left). Replacement of Leu$_{58}$ and Glu$_{59}$ with glycine (highlighted in red) is predicted to reduce the coiled coil probability to about 10% (lower panel, right). (**B**) The predicted coiled coil motif is critical for Lem8-mediated yeast toxicity. Yeast cells inducibly expressing Lem8 or mutant Lem8$_{GG}$ were serially diluted and spotted onto the indicated media for 48 hr (top panel). The expression of Lem8 and Lem8$_{GG}$ was examined and PGK1 was probed as a loading control (lower panel). (**C**) Lem8$_{GG}$ loses the capacity to cleave Phldb2 in mammalian cells. Lysates of HEK293T cells expressing Lem8 or Lem8$_{GG}$ were resolved by sodium dodecyl sulfate–polyacrylamide gel electrophoresis (SDS–PAGE) and detected by immunoblotting with antibodies specific for Phldb2 and HA, respectively. Tubulin was used as a loading control. Results shown were one representative from three independent experiments with similar results. (**D**) The predicted coiled coil motif is required for self-processing of Lem8. The indicated alleles of Lem8-GFP were individually expressed in HEK293T cells by transfection. Samples resolved by SDS–PAGE were detected by immunoblotting with GFP-specific antibodies. Results shown were one representative from three independent experiments with similar results. (**E**) Interactions between 14-3-3 ζ and the Lem8$_{GG}$ mutant. Lysates of 293T cells expressing Flag-14-3-3 ζ with HA-Lem8 or HA-Lem8$_{GG}$ were subjected to immunoprecipitation with the anti-HA antibody and the presence of 14-3-3 ζ in the precipitates was probed with the Flag-specific antibody. Results shown were one representative from three independent experiments with similar results.

to detect Lem8 activity against Phldb2 in infected cells is the low abundance or instability of the cleaved protein or a combination of both.

## A coiled coil motif in the amino terminal region of Lem8 is important for its interactions with 14-3-3ζ

Using the online MARCOIL sequence analysis software (*Gabler et al., 2020*), we identified a putative coiled coil motif located in the amino region of Lem8. Coiled coil is a common structural element in proteins, particularly those of eukaryotic origin; it is formed by two to seven supercoiled alpha-helices (*Liu et al., 2006*), and often is involved in protein–protein interactions, thus playing important roles in the formation of protein complexes (*Burkhard et al., 2001*). To determine the role of this region in the activity of Lem8, we introduced mutations to replace $Leu_{58}$ and $Glu_{59}$, the two sites predicted to be essential for the coiled coil structure in Lem8, with glycine (called $Lem8_{GG}$) (*Figure 5A*). When tested in yeast, these mutations have completely abolished the toxicity of Lem8 without affecting its expression or stability (*Figure 5B*). These mutations may affect the cysteine protease acitivity of Lem8, its interaction with the regulatory protein 14-3-3 ζ or its ability to recognize substrates.

We examined the ability of $Lem8_{GG}$ to cleave Phldb2 by coexpressing them in HEK293T cells. Whereas wild-type Lem8 consistently cleaves this substrate, $Lem8_{GG}$ has lost such activity despite a similar expression level (*Figure 5C*). To test the self-cleavage of $Lem8_{GG}$, we expressed Lem8-GFP or $Lem8_{GG}$-GFP in HEK293T cells and probed the fusion proteins by immunoblotting with GFP-specific antibodies. Comparing to Lem8-GFP, the levels of full-length proteins of Lem8-GFP and $Lem8_{C280S}$-GFP were similar, both were slightly higher than that of $Lem8_{GG}$-GFP and loss of the GFP portion of the protein occurred in the wild-type fusion but not in either of the two mutants (*Figure 5D*), indicating that the putative coiled coil domain is important for the self-cleavage activity of Lem8. Finally, we examined the impact of these mutations on the interaction between Lem8 and 14-3-3 ζ. Albeit $Lem8_{GG}$ expressed similar to the wild-type, it has largely lost the ability to bind 14-3-3 ζ in immuno-precipitation assays (*Figure 5E*). Together with the observation that Lem8 mutants lacking as few as 25 residues from its amino terminal end are unable to bind 14-3-3 ζ, these results suggest that the regulatory protein most likely binds Lem8 by recognizing the coiled coil motif located in its amino end region.

## Autocleaved Lem8 maintains the cysteine protease activity

It has been well established that some proteins, particularly enzymes are made as precursors or zymogens that need either autoprocessing or cleavage by other enzymes to exhibit their biological functions. One such example is caspases involved in cell death regulation and other important cellular functions. These enzymes are synthesized as zymogens before being activated by proteolytic cleavage in response to stimulation (*Shalini et al., 2015*). In some cases, autoprocessing leads to changes or even loss of their enzymatic activity (*Kapust et al., 2001*; *Zhang et al., 2018*). To investigate whether Lem8 that has undergone self-cleavage still possesses the cysteine protease activity, we tested the cleavage of Phldb2 by $Lem8_{\Delta C52}$, its self-processed form. Similar to full-length Lem8, $Lem8_{\Delta C52}$ was able to reduce the protein levels of Phldb2. In contrast, other truncation mutants, including $Lem8_{(\Delta)N25}$ and $Lem8_{\Delta N50}$ have lost the capacity to cleave Phldb2, while $Lem8_{\Delta C100}$ is poorly expressed in cells (*Figure 6A*). In addition, $Lem8_{\Delta C52}$, but not $Lem8_{\Delta N25}$ or $Lem8_{\Delta C100}$, cleaved the GFP tag from from the GFP-Phldb2 fusion and released the GFP signals from the plasma membrane (*Figure 6B*). Intriguingingly, although their ability to cleave Phldb2 appears similar, under our experimental conditions, the protein level of $Lem8_{\Delta C52}$ is considerably lower than that of Lem8 (*Figure 6A*), suggesting that the self-processed form has higher activity.

We next examined whether the protease activity of $Lem8_{\Delta C52}$ still requires 14-3-3 ζ binding. Results from immunoprecipitation and pulldown assays with purified protiens clearly showed that $Lem8_{\Delta C52}$ robustly binds 14-3-3 ζ (*Figure 6C, D*). Furthermore, incubation of $Lem8_{\Delta C52}$ with Phldb2 isolated from cells did not lead to its cleavage, but the inclusion of 14-3-3 ζ in this reaction led to cleavage (*Figure 6E*), indicating that $Lem8_{\Delta C52}$ still requires 14-3-3 ζ for its protease activity.

## Lem8 inhibits mammalian cell migration

Phldb2 is a PIP3-binding protein involved in microtubule stabilization (*Lansbergen et al., 2006*; *Paranavitane et al., 2003*), thus playing a pivotal role in cell motility. Depletion of Phldb2 significantly

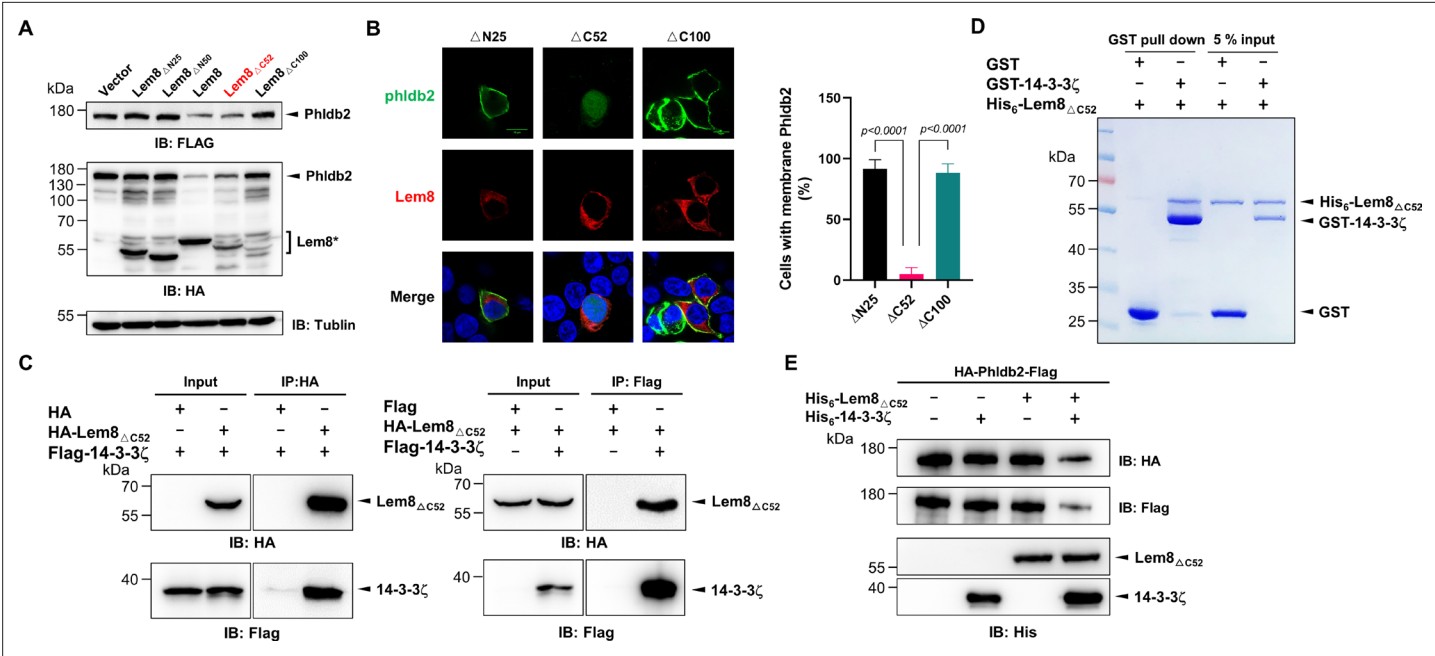

**Figure 6.** Autoprocessed Lem8 retains the cysteine protease activity. (**A**) The autoprocessed form of Lem8 cleaves Phldb2 in cells. HA-Phldb2-Flag was coexpressed in HEK293T cells with Lem8 or the indicated truncation mutants including the self-processed form, Lem8$_{\triangle C52}$. Twenty-four hours after transfection, the samples were resolved by sodium dodecyl sulfate–polyacrylamide gel electrophoresis (SDS–PAGE) and probed by a HA-specific antibody and a Flag-specific antibody. Tubulin was used as a loading control. Results shown were one representative from three independent experiments with similar results. (**B**) Lem8$_{\triangle C52}$ causes redistribution of GFP-Phldb2. Truncations of Lem8, including Lem8$_{\triangle N25}$, Lem8$_{\triangle C52}$, and Lem8$_{\triangle C100}$ fused to mCherry was individually expressed in HEK293T cells with GFP-Phldb2. Twenty-four hours after transfection, the fluorescence images were acquired with a Zeiss LSM 880 confocal microscope. The percentage of cells with membrane Phldb2 was calculated in Phldb2- and Lem8-positive cells (right panel). Bar, 10 µm. (**C**) The interaction between Lem8$_{\triangle C52}$ and 14-3-3 ζ . Total lysates of HEK293T cells transfected with indicated plasmid combinations were immunoprecipitated with antibodies specific for HA (left panel) or Flag (right), and the precipitates were probed with both HA and Flag antibodies. Similar results were obtained from at least three independent experiments and the data shown here were from one representative experiment. (**D**) Lem8$_{\triangle C52}$ directly interacts with 14-3-3 ζ . Mixtures containing GST-14-3-3 ζ and His$_6$-Lem8$_{\triangle C52}$ were incubated with glutathione beads for 1 hr at 4°C. After washing, samples resolved by SDS/PAGE were detected by Coomassie brilliant blue staining. Results shown were one representative from three independent experiments with similar results. (**E**) The cleavage of Phldb2 by Lem8$_{\triangle C52}$ requires 14-3-3 ζ . Purified HA-Phldb2-Flag from HEK293T was incubated with His$_6$-Lem8$_{\triangle C52}$ in reactions with or without His$_6$-14-3-3 ζ . Total proteins of all samples were resolved with SDS–PAGE, and probed by immunoblotting with antibody specific for HA, Flag, and His$_6$, respectively. Results shown were one representative from three independent experiments with similar results.

reduces the migration of MDA-231 cells in the haptotactic migration assay (*Astro et al., 2014*). As Lem8 cleaves Phldb2, we hypothesized that Lem8 may affect cell migration. To test this, we first established HEK293T-derived cell lines that stably express GFP, GFP-Lem8, or GFP-Lem8$_{C280S}$. Immunoblotting confirmed that Lem8 and Lem8$_{C280S}$ robustly expressed in the respective cell lines. Furthermore, in the cell line expressing Lem8, the level of Phldb2 was drastically reduced comparing to that in the line expressing GFP or Lem8$_{C280S}$ (*Figure 7A*). We then used the wound-healing scratch assay (*De Ieso and Pei, 2018*) to examine the impact of ectopic Lem8 expression on cell motility. Confluent monolayers of each cell line were scratched using a pipette tip and the migration of cells into the gap was monitored over a period of 24 hr. Results from this experiment showed that the percentage of wound closure at 24 hr after wounding was around 50% in samples using cells expressing GFP or GFP-Lem8$_{C280S}$. In the same experimental duration, cells expressing GFP-Lem8 only filled the gap by 26%, which was significantly slower than that of the controls (*Figure 7B*). Thus, ectopic expression of Lem8 inhibits mammalian cell migration.

An earlier study has shown that in the under-agarose migration assay, *L. pneumophila* inhibits the chemotaxis of mouse macrophages toward cytokines CCL5 and TNF-α in a Dot/Icm-dependent manner (*Simon et al., 2014*). Yet, the Dot/Icm substrates responsible for this inhibition remain elusive. We further studied whether Lem8 contributes to the inhibition of infected cell migration. To test this, we performed the scratch assay with 293T cells and Raw264.7 cells infected with relevant *L.*

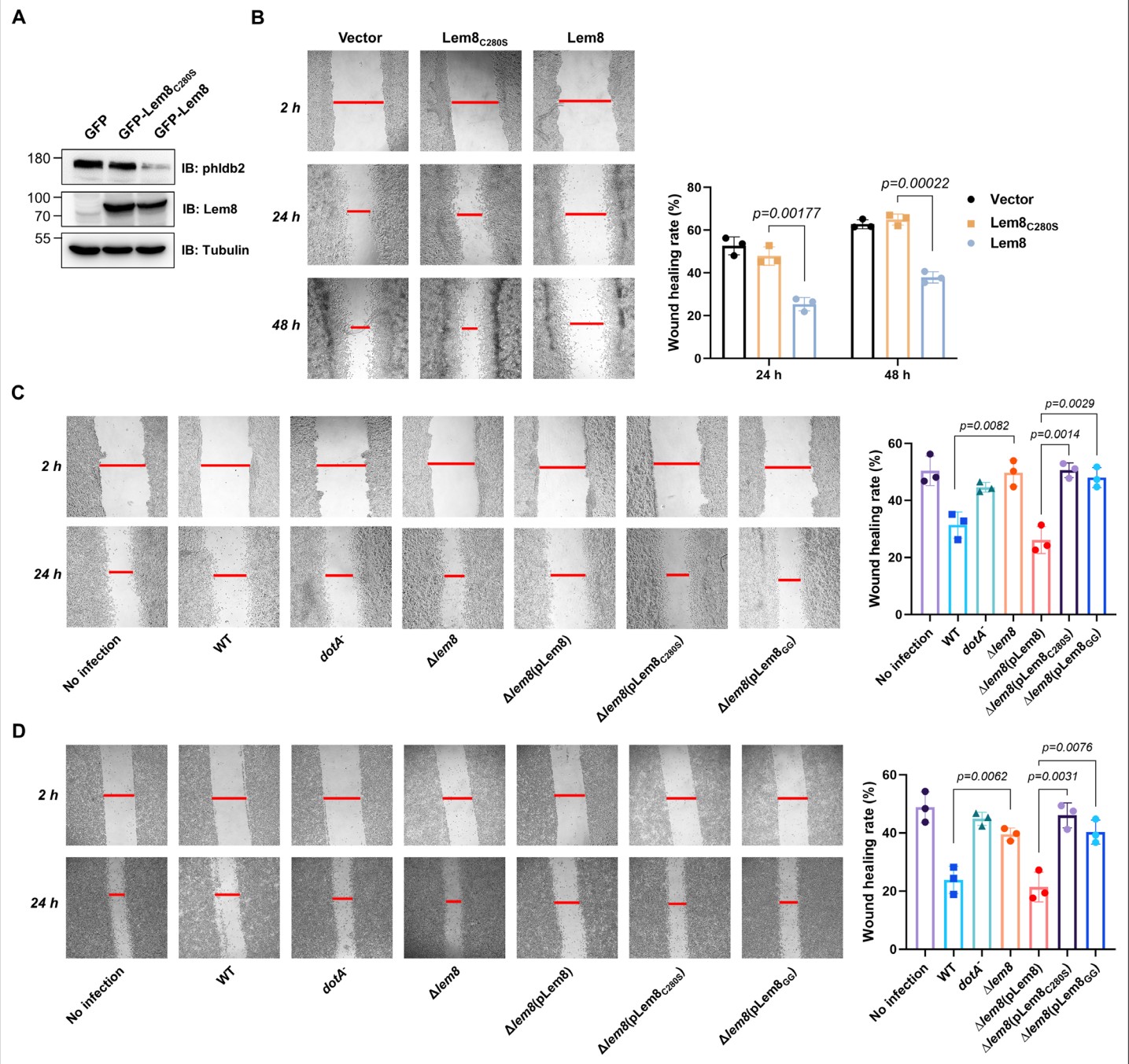

**Figure 7.** Lem8 contributes to cell migration inhibition by *L. pneumophila.* (**A**) Establishment of cell lines stably expressing Lem8 or its enzymatically inactive mutant. HEK293T cells were transduced with lentiviral particles harboring the indicated plasmid at an multiplicity of infection (MOI) of 10 for 2 days, and the GFP-positive cells were isolated by a BD Influx cell sorter. Lysates of each cell line were probed by immunoblotting with antibodies specific for Phldb2 or Lem8. Tubulin was used as a loading control. (**B**) Wound-healing scratch assay of the three stable cell lines. The three cell lines were individually seeded into 6-well plates. When reached confluency, cell monolayer of each cell line was scratched using a pipette tip. Images of the wounds were captured at 2, 24, and 48 hr after making the scratches using an Olympus IX-83 fluorescence microscope. Images of a representative experiment were shown (left panel). The wound-healing rates from three independent experiments were quantitated by Image J (right panel). (**C, D**) Evaluation of the impact of Lem8 on cell migration in cells infected with *L. pneumophila.* HEK293T cells expressing the FcγII receptor or Raw264.7 cells were infected with opsonized bacteria of the indicated *L. pneumophila* strains at an MOI of 50 for 2 hr. After washes, the wound-healing scratch assay was performed to evaluate the impact of infection on cell migration. Images of a representative experiment were shown (**C**, left panel) and the wound-healing rate was analyzed by Image J (**C**, right panel).

The online version of this article includes the following figure supplement(s) for figure 7:

**Figure supplement 1.** Overexpression of Phldb2 suppressed the inhibitory effects of Lem8 on cell migration.

*pneumophila* strains (*Figure 7C, D*). The percentage of wound closure by cells infected with wild-type *L. pneumophila* or Δ*lem8* (pLem8) was significantly lower than that with cells infected with the Δ*lem8* mutant. Consistent with its lack of the protease activity, Lem8$_{C280S}$ and Lem8$_{GG}$ were unable to complement the defects displayed by the Δ*lem8* mutant (*Figure 7C, D*). Thus, the inhibition of cell migration by *L. pneumophila* during infection is caused at least in part by the activity of Lem8.

To confirm the notion that Lem8 inhibits cell migration by attacking Phldb2, we performed the cell scratch assays using Lem8 expressing cells that had been transfected to ectopically express Phldb2. The level of Phldb2 in these cells considerably increased comparing to those transfected with a vector (*Figure 7—figure supplement 1A*). Importantly, the closure of the gap by cells overexpressing Phldb2 occurred significantly faster than those transfected with the vector (*Figure 7—figure supplement 1B*), validating our conclusion that Lem8 inhibits cell migration by attacking Phldb2.

## Discussion

Intracellular bacteria manipulate cellular processes to create a niche that supports their survival and replication in host cells by virulence factors that target proteins important for the regulation of these processes. These virulence factors often attack host regulatory proteins by diverse posttranslational modifications (PTMs) such as phosphorylation (*Krachler et al., 2011*), ubiquitination (*Zhou and Zhu, 2015*), AMPylation (*Yarbrough et al., 2009*), acetylation (*Mukherjee et al., 2006*), and ADP-ribosylation (*Cohen and Chang, 2018*). Proteolytic processing is a type of PTM that can lead to the activation, inactivation, or destruction of target proteins, causing alterations in cellular structure or signaling that benefit the pathogen. For instance, the type III effector EspL from enteropathogenic *E. coli* functions as a cysteine protease that antagonizes host inflammatory response by degrading several proteins involved in necroptotic signalling (*Pearson et al., 2017*). Our results herein establish Lem8 as a cysteine protease that directly targets the microtubule-associated protein Phldb2, therefore contributing to the inhibition of host cell migration by *L. pneumophila*. Lem8 joins a growing list of *Legionella* effectors with protease activity, including the serine protease Lpg1137 that inhibits autophagy by cleaving syntaxin 17 (*Arasaki et al., 2017*) and the metalloprotease RavK that attacks actin to disrupt the actin cytoskeleton of host cells (*Liu et al., 2017*).

One interesting feature associated with Lem8 is the requirement of 14-3-3 ζ for its activity. In line with the notion that amoebae are the primary host of *L. pneumophila*, the sole 14-3-3 protein from *D. discoideum* similarly activates Lem8. Due to extremely low bacterial uptake rates by *D. discoideum*, we were unable to examine the impact of Lem8 on the motility of this host. Nevertheless, despite Lem8 is not required for intracellular bacterial replication, it plays a role in *L. pneumophila* virulence by inhibiting host cell motility, this finding thus provides a molecular explanation for this phenotype observed earlier (*Simon et al., 2014*). It is worth noting that a close homolog of Phldb2 is not present in amoebae species such as *D. discoideum*. The fact that *lem8* is only present in a few isolates of the many sequenced *L. pneumophila* strains implies that these bacteria may have coevolved with higher eukaryotes. Several *L. pneumophila* Dot/Icm effectors that attacks host signaling mechanisms not known to exist in amoebae have been reported. For example, the activation of the NF-κB pathway by LegK1 (*Ge et al., 2009*) and the specific attack of linear polyubiquitin chains by RavD (*Wan et al., 2019*). These two signaing mechanisms are not present in lower eukaryotes such as amoebae (*Iwai, 2021*; *Li and Verma, 2002*). It is tempting to speculate that Lem8 has been acquired by strains of *L. pneumophila* that have coevolved with hosts that use Phldb2 or its close homologs to control cell motility.

In mammals, members of the 14-3-3 family, including 14-3-3 ζ often bind their client proteins by recognizing phosphorylated pockets with relatively conserved sequences such as RSX[pS/pT]XP (mode I) and RXXX[pS/pT]XP (mode II) (pS, phospho-serine, pT, phospho-threonine, X, any residue) (*Morrison, 2009*). Intriguingly, neither of these two motifs is present in Lem8. Consistently, recombinant Lem8 from mammalian cells or *E. coli* was not detectably phosphorylated (*Figure 2—figure supplement 1*). The binding of 14-3-3 proteins to unmodified clients is not unprecedented. All isoforms of 14-3-3 bind nonphosphorylated ExoS of *P. aeruginosa* by recongnizing the DALDL element (*Henriksson et al., 2002*), which bears sequence similarity to the unphosphorylated target WLDLE, an artificial R18 peptide inhibitor derived from a phage display library (*Petosa et al., 1998*). Elements with a sequence similar to these established recognition sites are not present in Lem8 nor is there one resembling those in other nonphosphorylated binding targets of 14-3-3 such as GPIb-α

(*Gu and Du, 1998*), p75NTR-associated cell death executor (NADE) (*Kimura et al., 2001*) or CLIC4 (*Suginta et al., 2001*).

Two lines of evidence suggest that 14-3-3 ζ recognizes a coiled coil motif in the amino terminal portion of Lem8. First, deletion of as few as 25 residues from the amino terminus end of Lem8 abolished its interaction with 14-3-3 ζ (*Figure 2D*). Second, the integrity of a predicted coiled coil motif in the amino terminal portion of Lem8 is required for its binding to the regulatory protein (*Figure 5*). Coiled coil motifs have long been known to be important for protein–protein interaction but its involvement in binding 14-3-3 has not yet been established. The binding of 14-3-3 to TRIM25 had been suggested to be mediated by recognizing a coiled coil domain, but the mechanism of such binding or whether phosphorylation is required remains unclear (*Gupta et al., 2019*). Future study, particularly structural analysis of the Lem8-14-3-3 ζ complex may allow a definite identification of the region in Lem8 recognized by 14-3-3 ζ, which will surely shed light on the additional features of the sequences recognizable by this imporant regulatory protein.

The self-cleavage of Lem8 has allowed us to identify its recognition sequence and several candidate cellular targets. One unexpected observation is that mutations in the identified recognition element reduced but did not abolish self-cleavage (*Figure 3D*). Thus, the primary sequence may not the only factor that dictates the specificity of Lem8 in substrate recognition. Other factors such as the overall structure of substrates may contribute to the configuration of the recognition site. The low level of specificity in cleavage site selection may allow Lem8 to more effectively bring down the cellular level of its targets, which may explain the requirement of 14-3-3 ζ for its activity. If a host cofactor is not needed for its activity, Lem8 may cleave itself or even other proteins in *L. pneumophila* cells. For Lem8, self-cleavage in the absence of 14-3-3 ζ in bacterial cells will be disastrous because the cleaved product will lose the portion of the protein that harbors translocation signals recognized by the Dot/Icm system (*Luo and Isberg, 2004*; *Nagai et al., 2005*). Likewise, the requirement of CaM by the Dot/Icm effector SidJ to inhibit the activity of members of the SidE family is to ensure that such inhibition does not occur in bacterial cells (*Bhogaraju et al., 2019*; *Black et al., 2019*; *Gan et al., 2019*; *Sulpizio et al., 2019*). The promiscuity in cleavage site recognition by Lem8 is also supported by the observation that this protease appears to cleave Phldb2 at multiple sites (*Figure 4*, *Figure 4—figure supplement 1*).

Interference with host cell motility appears to be a common strategy used by bacterial pathogens. For example, *S. enterica* Typhimurium inhibits the migration of infected macrophages and dendritic cells in a process that requires its type III effector SseI, which binds to IQGAP1, an important regulator of cell migration (*McLaughlin et al., 2009*). Similarly, the phosphatidylinositol phosphatase IpgD from *Shigella flexneri* contributes to the inhibition of chemokine-induced migration of human T cells (*Konradt et al., 2011*). The observation that cells infected with the wild-type *L. pneumophila* or strain Δ*lem8*(pLem8) migrated significantly slower than those infected with the Δ*lem8* mutant or its complementation strain expressing the Lem8$_{C280A}$ mutant suggests a role of Lem8 in cell mobility inhibition (*Simon et al., 2014*). Migration permits phagocytes including amoebae to move in response to chemotaxis cues, which likely will increase their chance to encounter cells that can recognize and engulf infected cells. Thus, cells with impaired motility are more likely to allow pathogens to complete their intracellular lifecycle.

Akin to most *L. pneumophila* Dot/Icm effectors, Lem8 is not required for proficient bacterial intracellular growth in commonly used laboratory hosts such as *D. discoideum*. Lem8 may be required for the survival of the bacteria in some specific inhabits or other Dot/Icm effectors may substitute its role by distinct mechanisms, thus contributing to such inhibition. Future studies aiming at the identification and characterization of Dot/Icm effectors involved in attacking host cells motility will continue to provide insights into the mechanisms of not only bacterial virulence but also the regulation of eukaryotic cell migration.

# Materials and methods

## Bacterial strains, plasmids, and cell culture

*E. coli* strain DH5α was used for plasmid construction and strain BL21 (DE3) or XL1blue was used for recombinant protein production and purification. All *E. coli* strains were grown on LB agar plates or in LB broth at 37°C. For maintenance of plasmids in *E. coli*, antibiotics were added in media at the

following concentrations: ampicillin, 100 µg/ml; kanamycin, 30 µg/ml. All *L. pneumophila* strains were derived from the Philadelphia one strain Lp02 and the *dotA⁻* mutant strain Lp03 (*Berger and Isberg, 1993*) and are listed in *Supplementary file 2*. *L. pneumophila* was cultured in N-(2-acetamido)-2-aminoethanesulfonic acid buffered yeast extract medium (AYE) or on charcoal buffered yeast extract plates (CYE). When necessary, thymidine was added into AYE at a final concentration of 0.2 mg/ml. pZL507 and its derivatives which allow expression of His$_6$-tagged proteins (*Xu et al., 2010*) in *L. pneumophila* were maintained by thymidine autotrophic. Deletion of the Lem8 coding gene lpg1290 (UniProtKB-Q5ZVZ8) from the genome of *L. pneumophila* was performed as described previously (*Liu and Luo, 2007*).

Plasmids used in this study are listed in *Supplementary file 2*. Genes were amplified by polymerase chain reactions (PCR) using Platinum SuperFi II Green PCR mix (Invitrogen, cat# 12369050). The PCR product was digested with restriction enzymes (New England Biolabs, NEB), followed by ligated to linearized plasmid using T4 DNA ligase (NEB). For site-directed mutagenesis, plasmid was reacted with primer pairs designed to introduce the desired mutations using Quikchange kit (Agilent, cat# 600670). After digestion with the restriction enzyme *DpnI* (NEB, cat# R0176), the products were transformed into the *E. coli* strain DH5α. All substitution mutants were verified by double strand DNA sequencing. The sequences of primers used for molecular cloning are listed in *Supplementary file 2*.

HEK293T and Hela cells purchased from ATCC were cultured in Dulbecco's modified minimal Eagle's medium supplemented with 10% fetal bovine serum (FBS). Bone marrow cells were isolated from 6- to 10-week-old female A/J mice (GemPharmatech, Co, Ltd) and were differentiated into BMDMs using L929-cell conditioned medium as described previously (*Conover et al., 2003*). PCR-based test (Sigma, cat# MP0025) was used to validate the absence of potential mycoplasma contamination in all mammalian cell lines. pAPH-HA, a plasmid suitable for expressing proteins with an amino HA tag and a carboxyl Flag tag (*Song et al., 2021*) was used to express tagged proteins in mammalian cells.

## β-Lactamase translocation assay

To test the Dot/Icm-dependent translocation into host cells, Lem8 was cloned intro pXDC61m (*Zhu et al., 2011*) to generate s β-lactamase-Lem8 fusion. This plasmid was introduced wild-type or the *dotA⁻* mutant of *L. pneumophila* and the resulting strains were used to infect U937 cells at an multiplicity of infection (MOI) of 20 after 0.5 mM isopropyl-β-D-thiogalactopyranoside (IPTG) induction. One hour after infection, the CCF4-AM substrates (Invitrogen, Carlsbad, CA) were added into the medium and the cells were incubated for another 2 hr at 25°C, followed by image acquisition using an Olympus IX-83 fluorescence microscope. The translocation of Lem8 was assessed by calculating the percentage of cells emitting blue fluorescence.

## Yeast manipulation

Unless otherwise indicated, yeast strains used in this study were derived from W303 (*Thomas and Rothstein, 1989*); yeast was grown at 30°C in yeast extract, peptone, dextrose medium (YPD) medium, or in appropriate amino acid dropout synthetic media supplemented with 2% of glucose or galactose as the sole carbon source.

For assessment of inducible protein toxicity, Lem8 or its derivatives were cloned into pYES2/NTA (Invitrogen) in which their expression is driven by the galactose-inducible promoter. Yeast transformation was performed using the lithium acetate method (*Gietz et al., 1995*). After growing in selective liquid medium with 2% raffinose, yeast cultures were serially diluted (fivefold) and 10 µl of each dilution was spotted onto selective plates containing glucose or galactose. Plates were incubated at 30°C for 3 days before image acquisition.

To screen Lem8-interacting protein(s), Gal4-based two-hybrid screening against the mouse cDNA library (Clontech) was performed as described before (*Mitsuzawa et al., 2005*). Briefly, Lem8$_{C280S}$ was inserted into pGBKT7 (*Banga et al., 2007*) to give pGBKLem8, which was transformed into the yeast strain PJ-64A (*James et al., 1996*) and the resulting strain was used for yeast two-hybrid screening. The mouse cDNA library was amplified in accordance with the manufacturer's instructions and the plasmid DNA was transformed into strain PJ-64A (pGBKLem8). Transformants were plated onto a selective synthetic medium lacking adenine, tryptophan, leucine, and histidine, colonies appeared on the selective medium were verified for interactions by reintroducing into strain PJ-64A (pGBKLem8)

and inserts of those that maintained the interaction phenotype were sequenced to identify the interacting proteins.

To validate the interactions between 14-3-3$\zeta$ and Lem8, its full-length gene was inserted into pGADGH (*Banga et al., 2007*) and the plasmids were introduced yeast strain PJ-64A (pGBKLem8). Yeast strains harboring the indicated plasmid combinations were streaked on Leu⁻ and Trp⁻ synthetic medium to select for plasmids and the transformants were transferred to Leu⁻, Trp⁻, Ade⁻, and His⁻ medium to examine protein–protein interactions measured by cell growth.

### Antibodies and immunoblotting

Polyclonal antibody against Lem8 were generated according to the protocol described before (*NCBI, 1996*; *Clark et al., 1997*). Briefly, 1 mg of emulsified His$_6$-Lem8 with complete Freund's adjuvant was injected intracutaneously into a rabbit four times at 10-day intervals. Sera of the immunized rabbit containing Lem8-specific antibodies were used for affinity purification of IgG with an established protocol (*Harlow, 1999*).

Samples from cells or bacterial lysates were prepared by adding 5× SDS loading buffer and heated at 95°C for 10 min. The soluble fraction of the lysates was resolved by SDS–PAGE, proteins were transferred onto polyvinylidene fluoride membranes (Pall Life Sciences). The membranes were blocking with 5% nonfat milk for 30 min, followed by incubated with primary antibodies at the indicated dilutions: α-Phldb2 (Sigma, cat# HPA035147, 1:1000), α-HA (Sigma, cat# H3663, 1:3000), α-Flag (Sigma, cat# F1804, 1: 3000), α-GFP (Proteintech, cat# 50430-2-AP, 1:5000), α-GST (Proteintech, cat# 66001-2, 1:10,000), α-His (Sigma, cat# H1029, 1: 3000), α-ICDH (1: 10,000) (*Xu et al., 2010*), α-Lem8 (1: 5000), α-PGK (Abcam, cat# ab113687, 1:2500), and α-Tubulin (Bioworld, cat# AP0064, 1:10,000). After washed three times, the membranes were incubated with appropriate HRP-labeled secondary antibodies and the signals were taken and analyzed by Tanon 5200 Chemiluminescent Imaging System.

### Transfection and immunoprecipitation

When grown to approximately 80% confluence, HEK293T cells were transfected using Lipofectamine 3000 (Invitrogen, cat# L3000150) according to the manufacturer's protocol. Twenty-four hours after transfection, cells were lysed using a lysis buffer (50 mM Tris–HCl, 150 mM NaCl, 0.5% Triton X-100, pH 7.5) for 10 min on ice, followed by centrifugation at 12,000 × *g* at 4°C for 10 min. Beads coated with Flag- (Sigma, cat# F2426), HA- (Sigma, cat# E6779), or GFP-specific antibodies (Sigma, cat# G6539) were washed twice with lysis buffer and then mixed with the prepared cell supernatant. The mixture was incubated on a rotatory shaker at 4°C overnight. The resin was washed with the lysis buffer for five times, followed by boiling in the Laemmli buffer at 95°C for 10 min to release the bound Flag- or HA-tagged proteins. For proteins used in biochemical reactions, the Flag- or HA-tagged proteins were eluted with Flag peptide (Sigma, cat# F4799) or HA peptide (Sigma, cat# I2149), respectively.

### Protein expression and purification

Lem8 and its mutants were amplified by PCR and cloned into pQE30 to express His$_6$-tagged proteins. The plasmids were transformed into in *E. coli* strain XL1blue and grown in LB broth. When the cell density reached an OD$_{600}$ of 0.8, IPTG was added into the cultures at a final concentration of 0.2 mM to induce the expression of target proteins for 14 hr at 16°C. Cells collected by centrifugation were resuspended in a lysis buffer (1× phosphate-buffered saline [PBS], 2 mM dithiothreitol [DTT], and 1 mM phenylmethylsulfonyl fluoride [PMSF]), and were lysed with a cell homogenizer (JN-mini, JNBIO, Guangzhou, China). The lysates were centrifugated at 20,000 × *g* for 30 min at 4°C twice to remove cell debris. The supernatant was incubated with Ni²⁺-NTA beads (QIAGEN) at 4°C for 1 hr, followed by washed with 50× bed volumes of 20 mM imidazole to remove unbound proteins. The His$_6$-tagged proteins were eluted with 250 mM imidazole in PBS buffer. Purified proteins were dialyzed in a storage buffer (30 mM NaCl, 20 mM Tris, 10% glycerol, pH 7.5) overnight at 4°C and then stored at −80°C.

14-3-3$\zeta$ and its homologous genes were cloned into pGEX6p-1 to express GST-tagged proteins. The plasmids were transformed into *E. coli* strain BL21 (DE3). Protein expression induction and purification were carried out similarly with Glutathione Sepharose 4B (GE Healthcare) beads. The resin was collected and washed for with wash buffer (lysis buffer plus 200 mM NaCl). The GST-tagged proteins were eluted with 10 mM glutathione and stored at −80°C after dialysis.

## In vitro cleavage assays

For autocleavage assays, 5 µg His$_6$-Lem8 or its mutants was incubated with or without 2.5 µg 14-3-3 ζ in 50 µl reaction buffer (50 mM Tris, 150 mM NaCl, pH 7.5) at room temperature for the indicated time points. For substrate candidates cleavage, Flag- or HA-tagged proteins purified from HEK293T cells were added into reactions with or without Lem8 and 14-3-3 ζ at room temperature for the indicated time. In each case, samples were analyzed by SDS–PAGE followed by immunoblotting or Coomassie brilliant blue staining.

## GST pulldown assay

GST-14-3-3 ζ or GST bound to Glutathione Sepharose 4B was incubated with His$_6$-Lem8 in a binding buffer (50 mM Tris, 137 mM NaCl, 13.7 mM KCl) for 2 hr at 4°C. After washing three times with the binding buffer, beads were boiled in the Laemmli buffer at 95°C for 10 min and the samples were resolved by SDS–PAGE. Proteins were detected by Coomassie brilliant blue staining.

## Bacterial infection, immunostaining, and image analysis

For infection experiments, *L. pneumophila* strains were grown in AYE broth to the postexponential growth phase (OD$_{600}$ = 3.3–3.8). When necessary, complementation strains were induced by 0.1 mM IPTG for another 4 hr at 37°C before infection.

To determine intracellular bacterial growth, *D. discoideum* or BMDMs of A/J mice were infected with relevant *L. pneumophila* at an MOI of 0.1. 2 hr after adding the bacteria, the cells were washed using warm PBS to remove the extracellular bacteria. *D. discoideum* and BMDMs were maintained in 22 and 37°C, respectively. At the indicated time points, cells were lysed with 0.2% saponin and appropriately diluted lysates were plated on CYE plates. After 4-day incubation at 37°C, the counts of bacterial colonies were calculated to evaluate the growth.

To determine the impact of the infection on cell migration, HEK293T cells transfected to express FcγRII receptor (*Qiu et al., 2016*) were infected with the indicated bacterial strains. Two hours after infection, cells were washed using warm PBS and were used for the wound-healing assay.

To determine the cellular localization of Lem8 in infected cells, BMDMs were infected with relevant *L. pneumophila* strains at an MOI of 10 for 2 hr. The samples were immunostained as described earlier (*Haenssler et al., 2015*). Briefly, we washed the samples three times with PBS to remove extracellular bacteria, and fixed the cells with 4% paraformaldehyde at room temperature for 10 min. After three times washes, cells were permeabilized using 0.1% Triton X-100 and then were blocked with 4% goat serum for 1 hr. Samples were incubated with rat anti-*Legionella* antibodies (1:10,000) and rabbit anti-Lem8 antibodies (1:100) at 4°C overnight followed by incubated with appropriate fluorescence-labeled secondary antibodies at room temperature for 1 hr. After stained by Hoechst 33,342 (Invitrogen, cat# H3570, 1:5000), samples were inspected using an Olympus IX-83 fluorescence microscope.

To detect the cleavage of endogenous Phldb2 by Lem8, Hela cells transfected to express mCherry-Lem8 or mCherry-Lem8$_{C280S}$ were stained with Phldb2-specific antibodies (1:100) as described above. The images were taken using a Zeiss LSM 880 confocal microscope. The determine the impact on ectopically expressed Phldb2, mCherry-Lem8, or mutants each was cotransfected with GFP-Phldb2 into HEK293T cells seeded onto glass coverslips (Nest, cat# 801001). Fixed samples were stained with Hoechst, cell images were acquired by a confocal microscope.

## Production of lentiviral particles and transduction

For production of lentiviral particles carrying *lem8* or its mutants, the *gfp-lem8* fusion was inserted into pCDH-CMV-MCS-EF1a-Puro (System Biosciences, cat# CD510B-1). The plasmids were cotransfected with pMD2.G (gift from Dr. Didier Trono, Addgene #12259) and psPAX2 (gift from Dr. Didier Trono, Addgene #12260) into HEK293T cells grown to about 70% confluence. Supernatant was collected after 48-hr incubation, followed by filtration with 0.45-µm syringe filters. After measuring the titers using qPCR with the Lentivirus Titer Kit (abm, cat# LV900), the packed lentiviral particles were used to infect newly prepared HEK293T cells at an MOI of 10. After incubation for 2 days, cells were sorted by BD Influx cell sorter to establish cell lines stably expressing the gene of interest.

## Mass spectrometry analysis of Lem8 self-cleavage site

Recombinant His$_6$-Lem8 was incubated with His$_6$-14-3-3 ζ for 8 hr and the samples were separated by SDS–PAGE. After Coomassie brilliant blue staining, bands corresponding full-length His$_6$-Lem8

or cleaved were excised and subjected to in-gel digestion with trypsin. Peptides were loaded into a nano-LC system (EASY-nLC 1200, Thermo Scientific) coupled to an LTQ-Orbitrap mass spectrometer (Orbitrap Velos, Thermo Scientific). Peptides were separated in a capillary column (75 μm × 15 cm) packed with C18 resin (Michrom BioResources Inc, 4 μm, 100 Å) with the following gradient: solvent B (100 ACN, 0.1% FA) was started at 7% for 3 min and gradually raised to 35% in 40 min, then rapidly increased to 90% in 2 min and maintained for 10 min before column equilibration with 100% solvent A (97% $H_2O$, 3% ACN, 0.1% FA). The flow rate was set at 300 nl/min and eluting peptides were directly analyzed in the mass spectrometer. Full-MS spectra were collected in the range of 350–1500 $m/z$ and the top 10 most intense parent ions were submitted to fragmentation in a data-dependent mode using collision-induced dissociation with the max injection time of 10 ms. MS/MS spectra were searched against the *L. pneumophila* (strain Philadelphia 1) database downloaded from UniProt using Mascot (Matrix Science Inc). The signals of Lem8 tryptic peptides were compared between full-length and cleaved samples to narrow down the potential cleavage site(s) within specific peptides, cleaved Lem8 semitryptic peptides were inspected manually.

## Wound-healing assay

Wound-healing assays were performed as previously described (*Liang et al., 2007*). Briefly, HEK293T cells or Raw264.7 cells were seeded into 6-well plates and incubated until the confluency reached about 90%. The cell monolayer was scraped in a straight line using a p200 pipet tip to create a 'wound', followed by washing with growth medium to remove the debris. Reduced-serum medium (1% serum) was added and the cells were placed back in a 37°C incubator. 2, 24, and 48 hr after making the scratch, images of the cell monolayer wound were taken using an Olympus IX-83 fluorescence microscope. For each image, distances between one side of the wound and the other were quantitated by Image J (http://rsb.info.nih.gov/ij/). The wound-healing rate was calculated by the following formula: % wound healing = (0 hr distance − 24 hr distance)/24 hr distance ×100.

## Data quantitation and statistical analyses

All data were represented as mean ± standard deviation. Student's *t*-test was applied to analyze the statistical difference between two groups each with at least three independent samples.

## Acknowledgements

The authors thank Dr. Shaohua Wang for plasmids, the study was funded in part by Jilin Science and Technology Agency grant 20200403117SF (LS), 20200901010SF (DL), National Natural Science Foundation of China grant 21974002 (XL), Beijing Municipal Natural Science Foundation grant 5202012 (XL), and the National Institutes of Health grant R01AI127465 (ZQL).

## Additional information

### Funding

| Funder | Grant reference number | Author |
| --- | --- | --- |
| Jilin Scientific and Technological Development Program | 20200403117SF | Lei Song<br>Lei Song |
| Jilin Scientific and Technological Development Program | 20200901010SF | Dan Li |
| National Natural Science Foundation of China | 21974002 | Xiaoyun Liu |
| Beijing Municipal Natural Science Foundation | 5202012 | Xiaoyun Liu |
| National Institutes of Health | R01AI127465 | Zhao-Qing Luo |

| Funder | Grant reference number | Author |
|--------|------------------------|--------|

The funders had no role in study design, data collection, and interpretation, or the decision to submit the work for publication.

## Author contributions
Lei Song, Conceptualization, Data curation, Project administration, Writing – original draft; Jingjing Luo, Dan Huang, Project administration; Hongou Wang, performed the mass spectrometric analysis; Yunhao Tan, Yao Liu, Conceptualization, Data curation, Project administration; Yingwu Wang, performed the mass spectrometric analysis; Kaiwen Yu, KY performed the mass spectrometric analysis; Yong Zhang, Methodology; Xiaoyun Liu, XL performed the mass spectrometric analysis; Dan Li, Conceptualization, Data curation, Writing – review and editing; Zhao-Qing Luo, Conceptualization, Data curation, Supervision, Writing – review and editing

## Author ORCIDs
Lei Song ⬥ http://orcid.org/0000-0002-4115-065X
Yao Liu ⬥ http://orcid.org/0000-0002-4330-2389
Kaiwen Yu ⬥ http://orcid.org/0000-0002-6050-8477
Zhao-Qing Luo ⬥ http://orcid.org/0000-0001-8890-6621

## Decision letter and Author response
Decision letter https://doi.org/10.7554/eLife.73220.sa1
Author response https://doi.org/10.7554/eLife.73220.sa2

# Additional files

## Supplementary files
• Transparent reporting form
• Supplementary file 1. Potential Lem8 interacting proteins identified by yeast two-hybrid screenings.
• Supplementary file 2. Bacterial strains, plasmids and primers used in the study.
• Source data 1. Source data for figures and figure supplements.

## Data availability
All data generated or analysed during this study are included in the manuscript. Files for original images of blots and gels prior to being cropped for use in the main text have been included in the Supporting file (zip format).

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
