## [Editor Report]

This manuscript provides new insight into the function of a Legionella pneumophila effector protein and indicate a novel mechanism by which the bacterial effector Lem8 interferes with host cell motility. These findings will be of broad interest to those in the bacterial pathogenesis and cytoskeleton fields, and represent an exciting advance in our understanding of host-pathogen interactions.

---

## [Decision Letter]

**Decision letter after peer review:**

Thank you for submitting your article "*Legionella pneumophila* regulates host cell motility by targeting Phldb2 with a 14-3-3ζ-dependent protease effector" for consideration by *eLife*. Your article has been reviewed by 3 peer reviewers, and the evaluation has been overseen by a Reviewing Editor and Jos Van der Meer as the Senior Editor. The reviewers have opted to remain anonymous.

Essential revisions:

(1) The presence of the co-factor 14-3-3ζ was shown to be essential for the protease activity of Lem8. However, the involvement of the co-factor in regulation of cell motility was not demonstrated and should be addressed experimentally.

(2) The authors could not detect any cleavage of Phldb2 during infection and the cleavage site(s) of Phldb2 remained to be unidentified, therefore, the link between loss of Phldb2 and motility suggested in the title "Legionella pneumophila regulates host cell motility by targeting Phldb2 with a 14-3-3ζ-dependent protease effector", is not experimentally established. Therefore, a number of other possible models exists, including the possibilities that unidentified substrates of Lem8 are involved in the phenotype or Lem8 is not a protease directly responsible for cleavage of Phldb2 and that the auto-cleaved Lem8 may acquire a distinctive enzymatic ability rather than a protease. To address testing their model that cleavage of Phldb2 by Lem8 affects motility, can the authors make a version of Phldb2 that is resistant to Lem8 cleavage? If so, expressing a Lem8-resistant version of Phldb2 in HEK293T cells may allow for the restoration of cell migration in cells expressing Lem8 or infected with Lem8-expressing Legionella. Alternatively, Lem8ΔC52 is a good substitute for the self-cleaved form of Lem8, showing the enhanced activity against Phlbd2 (Figure 6E). The authors may consider using this variant for the infection experiment. Transfecting the Lem8 stable-cell lines with Phldb2- expression plasmids could further clarify the role of the protein for this phenotype, although this does not directly test the role of cleavage. The authors may also examine the wound healing assay at a later timepoint.

(3) While the employed wound healing assay system is widely used, caution is needed when extrapolating the findings to more relevant host cells, amoeba or macrophages, without experimental validation. The observation that Lem8 is toxic for yeast, but not HEK293T cells highlights this. The text should reflect these limitations. Is Lem8 required for Legionella to interfere with motility of D. discoideum or other amoebae? Can the authors speculate in the discussion why it may be advantageous for Legionella to have an effector like Lem8 that may interfere with the migration of amoebae?

(4) The last sentence of the abstract states "Our results reveal a novel mechanism of inhibiting host cell motility by L. pneumophila for its virulence." However, the authors haven't shown a role for Lem8 in L. pneumophila virulence, as they show that Lem8 is dispensable for intracellular replication within D. discoideum and macrophages. The authors should modify this sentence and similar statements accordingly.

(5) As Figure 7 is the only and most important data that represents the biological significance of Lem8-Phldb2 axis for controlling cell migration, the images of the would-healing scratch plates should be presented more clearly (Panel C). For instance, WT vs Δlem8 does not seem to be so different in my eyes if the red lines are omitted, while the graph shows the two-fold difference. And the authors could include Lem8GG which lacks interaction with 14-3-3ζ, enabling to demonstrate the involvement of 14-3-3ζ in this axis.

(6) Line 144: The authors comment on the conservation Lem8 across Legionella spp., but not across L. pneumophila isolates. Is Lem8 conserved in all L. pneumophila isolates/strains?

(7) The authors confirm that Lem8 binds human and *Dictyostelium* 14-3-3 experimentally, but do not comment on evolutionary conservation of Phldb2. This should be at least discussed, as it is critical as to how far the findings from the study can be extrapolated.

(8) Figure 2A/ Y2H: The authors mention that many more clones apart from the ones containing 14-3-3 were identified in the Y2H screen. In the supplement, the authors should include a full list including information if additional interactions have been validated or disproven.

(9) Line 231: Do the authors mean Cleavage of C50 or C52 truncation mutant?

(10) Line 251: Figure 4: It is not clear if other candidate proteins apart from Phldb2 were tested. If yes and they were not cleaved, this information should be included to avoid unnecessary duplication of experiments by other researchers.

(11) In Figure S5C, Cleavage occurs with the HA-Lem8C280S mutant, instead of WT? Were the figure labels swapped?

(12) Please include accession numbers Uniprot/ NCBI WP accession numbers for Lem8 in the material and methods section for cross referencing.

(13) In figure 4D, the authors show microscopy images indicating that in cells expressing WT Lem8, the GFP signal is redistributed throughout the cytoplasm. The authors should quantify this data in a large number of cells and depict it as a graph alongside the microscopy images. Similarly, the authors should show graphs depicting quantification alongside representative microscopy images shown in Figure S5A.

(14) In figure 6A, Lem8 deltaC100 is poorly expressed, and thus it is difficult to draw any definitive conclusions from this mutant. The authors should clarify this point in their manuscript.

(15) Figure 7C: There are two sets of microscopy images labeled deltalem8 (Lem8). Was one of the sets meant to represent infection with deltalem8 Legionella complemented with the Lem8C280S mutant?

(16) In the figure legends, the authors mention how many times experiments were performed for some figures, but not for others. Please state for each figure how many times each experiment was performed.

(17) Contributions from one of the corresponding authors (DL) was not described in "Author contributions" section. Please specify the author's contributions. ("SL" is supposed to be "LS".)

(18) The authors could consider analyzing the cleaved Lem8 (4A) (Figure 3D) by MS to examine if the same site (476) was cleaved or not.

(19) Figure 3B should include the control without 14-3-3ζ.

(20) The β-lactamase translocation assay (Figure 1B) should be described in Materials and methods.

(21) Figure S2A and S5A: The scale bar is missing.

(22) Figure S5B the positions of the arrowheads are not accurate.

(23) Are the N-terminal HA-tagged Lem8 derivatives pAPH-derived, please indicate the information in Table S1.

(24) Lines 180-183: No data was presented for ExoS in any figures.

(25) Lines 215-217: Could not find the A468PQPTPQRQ476- fragment in Figure 3C.

(26) Line 266: The location of the PIP3-binding domain in Phldb2 should be described (if it is the C-terminus or not).

(27) Lines 299-300: No data about Phldb2ΔC153 in Figure S5C.

(28) Lines 328-329: The statement does not accurately describe the data in Figure 5D.

---

## [Author Response]

Essential revisions:(1) The presence of the co-factor 14-3-3ζ was shown to be essential for the protease activity of Lem8. However, the involvement of the co-factor in regulation of cell motility was not demonstrated and should be addressed experimentally.

We agree that probing the role of 14-3-3ζ in Lem8-induced cell motility inhibition will further support the conclusion that it functions as a co-factor for the effector. Determining such role in cells would need to interfere with 14-3-3ζ expression. Yet, numerous studies have demonstated that knockdown of 14-3-3ζ alone impairs cell motility (e.g: Chen, Liu et al., 2014, Goc, Abdalla et al., 2012, Jin, Han et al., 2016, Kambach, Sodi et al., 2014, Kim, Ullevig et al., 2014, Kobayashi, Ogura et al., 2011, Li, Xu et al., 2019, Lu, Guo et al., 2009, Preisinger, Short et al., 2004, Yu, Chen et al., 2017). The fact that 14-3-3ζ itself is required for cell motifity would prevent us from making any conclusion on its role in promoting the cell motility-inhibition activity of Lem8. We would like to note that our biochemical experiments have unequivocally shown that Lem8 requires 14-3-3ζ for cleaving itself and its host target Phldb2 (Figures 3 and 4).

(2) The authors could not detect any cleavage of Phldb2 during infection and the cleavage site(s) of Phldb2 remained to be unidentified, therefore, the link between loss of Phldb2 and motility suggested in the title "Legionella pneumophila regulates host cell motility by targeting Phldb2 with a 14-3-3ζ-dependent protease effector", is not experimentally established. Therefore, a number of other possible models exists, including the possibilities that unidentified substrates of Lem8 are involved in the phenotype or Lem8 is not a protease directly responsible for cleavage of Phldb2 and that the auto-cleaved Lem8 may acquire a distinctive enzymatic ability rather than a protease. To address testing their model that cleavage of Phldb2 by Lem8 affects motility, can the authors make a version of Phldb2 that is resistant to Lem8 cleavage? If so, expressing a Lem8-resistant version of Phldb2 in HEK293T cells may allow for the restoration of cell migration in cells expressing Lem8 or infected with Lem8-expressing Legionella. Alternatively, Lem8ΔC52 is a good substitute for the self-cleaved form of Lem8, showing the enhanced activity against Phlbd2 (Figure 6E). The authors may consider using this variant for the infection experiment. Transfecting the Lem8 stable-cell lines with Phldb2- expression plasmids could further clarify the role of the protein for this phenotype, although this does not directly test the role of cleavage. The authors may also examine the wound healing assay at a later timepoint.

We appreciate the vigor of the reviewer and her/his insightful suggestions. Our data indicate that the cleavage of Phldb2 by Lem8 occurs at multiple sites. Truncated protein fragments are known to be susceptible to nonspecific digestions by proteases, which likely accounts for our inability to detect Phldb2 fragments produced by Lem8.

We agree that detection of biochemical effects imposed by bacterial effectors under infection conditions will provide a strong support of the conclusion but such effects in some scenarios are extremely challenging and the detection of Lem8-mediated cleavage of Phldb2 in cells infected by *L. pneumophila* clearly is one such case. This is one of the most difficult parts of projects on functional analysis of bacterial effectors. As we have discussed in our manuscript, this can be due to a combination of scarce amounts of translocated Lem8, the high abundance of cellular Phldb2 and the inability to detect the cleaved products due to the presence of multiple cleavage sites. Although it is formally possible that Lem8 has an alternative enzymatic activity beyond the protease activity describe here. Our cleavage experiments with recombinant proteins (Figure 4) and results from new experiments showing that overexpression of Phldb2 suppressed the migration inhibition imposed by Lem8 (Figure 7 figure supplement 1) strongly support the conclusion that attack of Phldb2 by Lem8 led to inhibition of cell migration.

That fact that Lem8 cleaves Phldb2 at multiple sites makes it extremely difficult if not impossible to make mutants that are resistant to the effector. Instead, we have followed the suggestion to overexpress Phldb2 in cells transfected to express Lem8 and found that it indeed can suppress the motility inhibition imposed by Lem8 (Figure 7 figure supplement 1). These results have strengthened our conclusion that Lem8 inhibits host cell motility by attacking Phldb2.. We also have extended the duration of the experiments by adding a 48-h time point (Figures 7B and Figure 7 figure supplement 1). Please note that we did not include the data of 48-h under infection condition because the cells became very unhealthy at this time point.

As for the suggestion to express the Lem8ΔC52 mutant in *L. pneumophila* and test its impact on cell motility. It has been well-established that signals required for Dot/Icm-mediated protein translocation resides in the carboxyl end of the effectors (Lifshitz, Burstein et al., 2013, Nagai, Cambronne et al., 2005). As a result, Lem8ΔC52 cannot be recognized and translocated by Dot/Icm, which makes the experiment technically impossible to perform. We have speculated that one of the possible reasons for the requirement of 14-3-3ζ is to ensure that self-cleavage of Lem8 does NOT occur within bacterial cells (page 17).

(3) While the employed wound healing assay system is widely used, caution is needed when extrapolating the findings to more relevant host cells, amoeba or macrophages, without experimental validation. The observation that Lem8 is toxic for yeast, but not HEK293T cells highlights this. The text should reflect these limitations. Is Lem8 required for Legionella to interfere with motility of D. discoideum or other amoebae? Can the authors speculate in the discussion why it may be advantageous for Legionella to have an effector like Lem8 that may interfere with the migration of amoebae?

We would like to thank the reviewer for raising this important point. We have attempted the experiments using the amoebae host *Dictyostelium discoideum*. Unfortunately, the rate of bacterial uptake by this host is extremely low, which prevents us from making any definite conclusion. In the same time, experiments with Raw264.7 macrophages showed that Lem8 plays a role in the inhibition of cell motility during *L. pneumophila* infection (revised Figure 7D). Finally, we have followed the suggestion to discuss these results in the context of infection of more “natural” hosts of *L. pneumophila* by noting the limitations of the experimentals systems we used (page 15).

We have added a brief discussion of the potential advantage for the *L. pneumophila* to inhibit host cell migration (page 18).

(4) The last sentence of the abstract states "Our results reveal a novel mechanism of inhibiting host cell motility by L. pneumophila for its virulence." However, the authors haven't shown a role for Lem8 in L. pneumophila virulence, as they show that Lem8 is dispensable for intracellular replication within D. discoideum and macrophages. The authors should modify this sentence and similar statements accordingly.

We have revised the text to reflect the fact that Lem8 is not required for optimal intracellular replication (page 2 Abstract, page 15 bottom).

(5) As Figure 7 is the only and most important data that represents the biological significance of Lem8-Phldb2 axis for controlling cell migration, the images of the would-healing scratch plates should be presented more clearly (Panel C). For instance, WT vs Δlem8 does not seem to be so different in my eyes if the red lines are omitted, while the graph shows the two-fold difference. And the authors could include Lem8GG which lacks interaction with 14-3-3ζ, enabling to demonstrate the involvement of 14-3-3ζ in this axis.

We have followed the suggestion to redo the experiment with additional samples that express LemGG. The results showed that this mutant has lost the ability to inhibit cell motility (revised Figure 7C), which further validates our conclusion that 14-3-3ζ-induced Lem8 activity is responsible for the phenotype.

(6) Line 144: The authors comment on the conservation Lem8 across Legionella spp., but not across L. pneumophila isolates. Is Lem8 conserved in all L. pneumophila isolates/strains?

We have further explored the distribution of Lem8 in L. pneumophila isolates and strains and found that it is present in the following strains. In addition, we have added a discussion point to reflect this finding (Page 6, Line 144 to Line148).

**Author response image 1. sa2fig1:** 

Lem8 is conserved insubsp. pneumophila ATCC 43290 (serogroup 12)

subsp. pneumophila Thunder Bay

subsp. pneumophila LPE509

Paris (serogroup 1)

subsp. pneumophila HL06041035 (serogroup 1)

We have added this information in the revised text and cited the corresponding references (page 6).

(7) The authors confirm that Lem8 binds human and *Dictyostelium* 14-3-3 experimentally, but do not comment on evolutionary conservation of Phldb2. This should be at least discussed, as it is critical as to how far the findings from the study can be extrapolated.

Thanks for raising this interesting point! We have further searched for potential Phldb2 ortholog in *D. discoideum* and found that the protein coded for by gene DDB_G0288073 bears remote similarity (please see sequence alignment Author response image 2). We have further dicussed the implications of targeting Phldb2, a protein only present in mammalian cells or other higher organisms (page 15-16).

As a point of discussion in this context, it is worth noting that the scientific community of Legionella biology has long believes that unicellular eukaryotes in fresh water niches provide the main driving force for the evolution of these bacteria. Whereas this notion is still largely correct, over the years, the activity of several of its Dot/Icm effectors suggests that these bacteria may have co-evolved with higher organisms. For example, RavD specifically attacks linear ubiquitina chains, a strong inducer for immunity that so far has only been found in mammalian cells (Iwai, 2021, Wan, Wang et al., 2019), so was the kinase LegK1 that activate the noncannical branch of the NFκB pathway (Ge, Xu et al., 2009), which does not exist in amoeba. We have briefly discussed this point in the revised manuscript (page 16).

(8) Figure 2A/ Y2H: The authors mention that many more clones apart from the ones containing 14-3-3 were identified in the Y2H screen. In the supplement, the authors should include a full list including information if additional interactions have been validated or disproven.

We have included a supplementary Table (Table S1) showing other genes obtained in the Y2H screening.

(9) Line 231: Do the authors mean Cleavage of C50 or C52 truncation mutant?

Thanks for spotting this typo. It should be C52 truncation. We have made a correction.

(10) Line 251: Figure 4: It is not clear if other candidate proteins apart from Phldb2 were tested. If yes and they were not cleaved, this information should be included to avoid unnecessary duplication of experiments by other researchers.

Our results showed that Lem8 is unable to cleave the other candidates. We have included the results in revised Figure 4 figure supplement 1 A.

(11) In Figure S5C, Cleavage occurs with the HA-Lem8C280S mutant, instead of WT? Were the figure labels swapped?

This is a typo (has been corrected in S5D)

(12) Please include accession numbers Uniprot/ NCBI WP accession numbers for Lem8 in the material and methods section for cross referencing.

Information added as requested.

(13) In figure 4D, the authors show microscopy images indicating that in cells expressing WT Lem8, the GFP signal is redistributed throughout the cytoplasm. The authors should quantify this data in a large number of cells and depict it as a graph alongside the microscopy images. Similarly, the authors should show graphs depicting quantification alongside representative microscopy images shown in Figure S5A.

We have performed the analysis as suggested. The results are shown in revised Figure 4D. Similar analysis has been done for Figure S5A (now Figure 4 figure supplement 1B).

(14) In figure 6A, Lem8 deltaC100 is poorly expressed, and thus it is difficult to draw any definitive conclusions from this mutant. The authors should clarify this point in their manuscript.

Thank you. We have added a note to reflect this point in the text (Page 13).

(15) Figure 7C: There are two sets of microscopy images labeled deltalem8 (Lem8). Was one of the sets meant to represent infection with deltalem8 Legionella complemented with the Lem8C280S mutant?

This is an error in our part. Thank you for pointing it out. We have more carefully inspected the images and presented them correctly.

(16) In the figure legends, the authors mention how many times experiments were performed for some figures, but not for others. Please state for each figure how many times each experiment was performed.

Thank you! we have revised the legends to reflect the repeats we have done for all experiments.

(17) Contributions from one of the corresponding authors (DL) was not described in "Author contributions" section. Please specify the author's contributions. ("SL" is supposed to be "LS".)

Thank you for pointing this out. We have corrected these errors.

(18) The authors could consider analyzing the cleaved Lem8 (4A) (Figure 3D) by MS to examine if the same site (476) was cleaved or not.

Thank you. We have determined the site of cleavage in the Lem8(4A) mutant, which is 9 residues upstream of the site identified in wild-type protein (Figure 3 figure supplement 1C). We have described the results in page 9.

(19) Figure 3B should include the control without 14-3-3ζ.

We have added images from samples without 14-3-3ζ.

(20) The β-lactamase translocation assay (Figure 1B) should be described in Materials and methods.

We have followed the suggestion to add the experimental procedure in the Methods section (page 23).

(21) Figure S2A and S5A: The scale bar is missing.

Thank you! we have added scal bars to these panels.

(22) Figure S5B the positions of the arrowheads are not accurate.

Thank you! corrected.

(23) Are the N-terminal HA-tagged Lem8 derivatives pAPH-derived, please indicate the information in Table S1.

Thank you! the feature of the plasmid pAPH has recently published (Song, Xie et al., 2021), we have added the information in Table S2 and cited the reference.

(24) Lines 180-183: No data was presented for ExoS in any figures.

We have revised the text to accurately reflect the resuls in Figure S3 (now Figure 2 figure supplement 1).

(25) Lines 215-217: Could not find the A468PQPTPQRQ476- fragment in Figure 3C.

The fragment should be L_464_CEKAPQPTPQRQ_476._ We have corrected the error in the revised text.

(26) Line 266: The location of the PIP3-binding domain in Phldb2 should be described (if it is the C-terminus or not).

We have added the information that the PIP3-binding motif PH domain of PHldb2 is localized to its carboxyl end.

(27) Lines 299-300: No data about Phldb2ΔC153 in Figure S5C.

We have now included results from both ∆C153 and ∆C100.

(28) Lines 328-329: The statement does not accurately describe the data in Figure 5D.

Thank you for pointing out this. We have carefully revised the text to more accurately describe the results (page 12).

References:

Chen M, Liu T, Xu L, Gao X, Liu X, Wang C, He Q, Zhang G, Liu L (2014) Direct interaction of 14-3-3zeta with ezrin promotes cell migration by regulating the formation of membrane ruffle. *J Mol Biol* 426: 3118-3133

Ge J, Xu H, Li T, Zhou Y, Zhang Z, Li S, Liu L, Shao F (2009) A Legionella type IV effector activates the NF-kappaB pathway by phosphorylating the IkappaB family of inhibitors. *Proc Natl Acad Sci U S A* 106: 13725-30

Goc A, Abdalla M, Al-Azayzih A, Somanath PR (2012) Rac1 activation driven by 14-3-3zeta dimerization promotes prostate cancer cell-matrix interactions, motility and transendothelial migration. *PLoS One* 7: e40594

Iwai K (2021) Discovery of linear ubiquitination, a crucial regulator for immune signaling and cell death. *FEBS J* 288: 1060-1069

Jin LM, Han XH, Jie YQ, Meng SS (2016) 14-3-3zeta silencing retards tongue squamous cell carcinoma progression by inhibiting cell survival and migration. *Cancer Gene Ther* 23: 206-13

Kambach DM, Sodi VL, Lelkes PI, Azizkhan-Clifford J, Reginato MJ (2014) ErbB2, FoxM1 and 14-3-3zeta prime breast cancer cells for invasion in response to ionizing radiation. *Oncogene* 33: 589-98

Kim HS, Ullevig SL, Nguyen HN, Vanegas D, Asmis R (2014) Redox regulation of 14-3-3zeta controls monocyte migration. *Arterioscler Thromb Vasc Biol* 34: 1514-21

Kobayashi H, Ogura Y, Sawada M, Nakayama R, Takano K, Minato Y, Takemoto Y, Tashiro E, Watanabe H, Imoto M (2011) Involvement of 14-3-3 proteins in the second epidermal growth factor-induced wave of Rac1 activation in the process of cell migration. *J Biol Chem* 286: 39259-68

Li J, Xu H, Wang Q, Wang S, Xiong N (2019) 14-3-3zeta promotes gliomas cells invasion by regulating Snail through the PI3K/AKT signaling. *Cancer Med* 8: 783-794

Lifshitz Z, Burstein D, Peeri M, Zusman T, Schwartz K, Shuman HA, Pupko T, Segal G (2013) Computational modeling and experimental validation of the Legionella and Coxiella virulence-related type-IVB secretion signal. *Proc Natl Acad Sci U S A* 110: E707-15

Lu J, Guo H, Treekitkarnmongkol W, Li P, Zhang J, Shi B, Ling C, Zhou X, Chen T, Chiao PJ, Feng X, Seewaldt VL, Muller WJ, Sahin A, Hung MC, Yu D (2009) 14-3-3zeta Cooperates with ErbB2 to promote ductal carcinoma in situ progression to invasive breast cancer by inducing epithelial-mesenchymal transition. *Cancer Cell* 16: 195-207

Nagai H, Cambronne ED, Kagan JC, Amor JC, Kahn RA, Roy CR (2005) A C-terminal translocation signal required for Dot/Icm-dependent delivery of the Legionella RalF protein to host cells. *Proc Natl Acad Sci U S A* 102: 826-31

Preisinger C, Short B, De Corte V, Bruyneel E, Haas A, Kopajtich R, Gettemans J, Barr FA (2004) YSK1 is activated by the Golgi matrix protein GM130 and plays a role in cell migration through its substrate 14-3-3zeta. *J Cell Biol* 164: 1009-20

Song L, Xie Y, Li C, Wang L, He C, Zhang Y, Yuan J, Luo J, Liu X, Xiu Y, Li H, Gritsenko M, Nakayasu ES, Feng Y, Luo ZQ (2021) The Legionella Effector SdjA Is a Bifunctional Enzyme That Distinctly Regulates Phosphoribosyl Ubiquitination. *mBio*: e0231621

Wan M, Wang X, Huang C, Xu D, Wang Z, Zhou Y, Zhu Y (2019) A bacterial effector deubiquitinase specifically hydrolyses linear ubiquitin chains to inhibit host inflammatory signalling. *Nat Microbiol* 4: 1282-1293

Yu J, Chen L, Chen Y, Hasan MK, Ghia EM, Zhang L, Wu R, Rassenti LZ, Widhopf GF, Shen Z, Briggs SP, Kipps TJ (2017) Wnt5a induces ROR1 to associate with 14-3-3zeta for enhanced chemotaxis and proliferation of chronic lymphocytic leukemia cells. *Leukemia* 31: 2608-2614